
# A conflicts' classification for IoT-based services: a comparative survey

Hamada Ibrhim[1,*], Hesham Hassan[2] and Emad Nabil[2,3,*]

[1] Computer Science Department, Faculty of Computers and Information, Minia University, Minia, Egypt
[2] Computer Science Department, Faculty of Computers and Artificial Intelligence, Cairo University, Giza, Egypt
[3] Computer Science Department, Faculty of Computer and Information Systems, Islamic University in Madinah, Madinah, Saudi Arabia
[*] These authors contributed equally to this work.

## ABSTRACT

Recently, Internet of Things (IoT)-based systems, especially automation systems, have become an indispensable part of modern-day lives to support the controlling of the networked devices and providing context-aware and intelligent environments. IoT-based services/apps developed by the end-users interact with each other and share concurrent access to devices according to their preferences, which increases safety, security, and correctness issues in IoT systems. Due to the critical impacts resulting from these issues, IoT-based apps require a customized type of compilers or checking tools that capable of analyzing the structures of these apps and detecting different types of errors and conflicts either in intra-IoT app instructions or in inter-IoT apps interactions. A plethora of approaches and frameworks have been proposed to assist the best practices for end-users in developing their IoT-based apps and mitigate these errors and conflicts. This paper focuses on conflict classification and detection approaches in the context of IoT systems by investigating the current research techniques that provided conflicts' classification or detection in IoT systems (published between 2014 and 2020). A classification of IoT-based apps interaction conflicts is proposed. The proposed conflicts' classification provides a priori conflicts detection method based on the analysis of IoT app instructions' relationships with utilizing the state-of-the-art Satisfiability Modulo Theories (SMT) model checking and formal notations. The current detection approaches are compared with each other according to the proposed conflicts' classification to determine to which extend they cover different conflicts. Based on this comparison, we provide evidence that the existing approaches have a gap in covering different conflicts' levels and types which yields to minimize the correctness and safety of IoT systems. We point out the need to develop a safety and security compiler or tool for IoT systems. Also, we recommend using a hybrid approach that combines model checking with a variety of languages and semantic technologies in developing future IoT-based apps verification frameworks to cover all levels and types of conflicts to guarantee and increase the safety, security, and correctness of IoT systems.

Corresponding author
Hesham Hassan, h.hassan@fci-cu.edu.eg

## INTRODUCTION

Automation of large and complex buildings such as houses, hospitals, universities, and other commercial buildings requires a multi-purpose system that can perform different tasks. This type of system is capable of supporting end-users with a user-friendly GUI to develop their automation tasks such as: detecting bugs and errors in these apps, distinguishing conflicts in apps' interactions, orchestrating apps, and providing data integration over-attached devices. The main features of these multi-purpose systems are accomplishing tasks automatically and providing different facilities in the building based on users' preferences. These facilities range from predicting user occupancy and turn on lights to HVAC, security, fire-alarm, maintaining comfortable utility services for occupants, and combining several vendor-based systems to achieve the required goals. In IoT-based systems, the embedded sensors and actuators offer their functionalities via service-oriented interfaces such as SOAP-based (web services) (*Teixeira et al., 2011*) or RESTful (*Pautasso, Zimmermann & Leymann, 2008*).

To the best of our knowledge, we can categorize Internet of Things (IoT) automation facilities into two main categories, "An Automation Service" and "A System Policy". There is an abundance of definitions of IoT automation services and system policies. In this context, we will define them in the following ways. An Automation Service (later referred to as 'IoT app' or 'service' interchangeably) is a set of RULES authored by an end-user and to personalize the actions of specific devices in a spatio-temporal of interest. Each service rule is a set of activating conditions (based on device outputs or environment state change) and a set of actions to perform. An example of automation service is to open Fan and Window of a lecture hall when the number of students is greater than 60.[1] A System Policy (later referred to as 'policy') is a set of CONSTRAINTS and obligations used to specify acceptable ranges for a set of devices. System policies are created by system admins (or rarely by end-users) to ensure that a smart IoT system meets the intended needs of its users and to limit the actions that may cause harm for either the occupants, the IoT environment, or the building itself. An example of environmental policy is to ensure that the lecture hall AC is turned off when no one is in the hall. System policies have been used in previous works with different synonyms as *Requirement Dependencies* (*Munir & Stankovic, 2014*), *Properties* (*Zhang et al., 2019a*), *Constraints* (*Le Guilly et al., 2016*), and *Devices Communication policies* (*Nagendra et al., 2019*). On the other hand, they used policies differently to resolve the conflicts between different users.

Automation services and system policies can be represented using different notation styles of automation programming tools and platforms in particular Rule-based tools (*Shafti et al., 2013*) such as IFTTT (*IFTTT, 2020*), Tasker (*Tasker, 2021*), Atooma (*Atooma, 2021*) and Trigger-Action programming (TAP) (*Ur et al., 2014*). These Rule-based tools are considered the most popular representations for IoT apps, due to their expressiveness and simplicity as depicted in Fig. 1.

User-defined automation services may be partially or fully realized by the system depending on system policies and other users' automation services affecting the same spaces. Partial realization results from the following reasons (among others): (1) increasing

[1] This can be detected and counted using, for example, passive infrared (PIR) detectors or video cameras

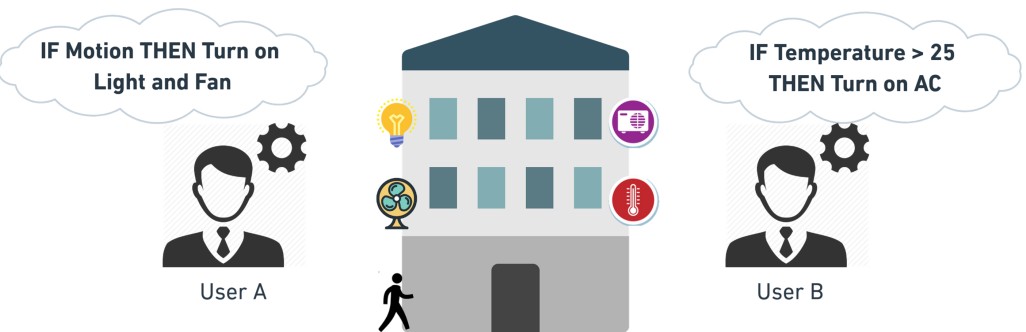

**Figure 1   Example of building environment control via two users.**

number of authored services from non-technical users, which characterized by different users preferences, (2) potential authoring errors in automation services, (3) unexpected hardware and software bugs, (4) increasing number of heterogeneous resources (e.g., devices), and (5) It's quite difficult for developers of IoT systems to satisfy all users' preferences when developing such systems. IoT apps programming bugs and conflicts in intra-interactions and inter-interactions between users' IoT apps are the main results of these reasons.

Previous researches have shown such different categories of conflicts affecting IoT system correctness and safety (*Delicato et al., 2013*; *Zhang et al., 2018*; *Alharithi, 2019*; *Huang, Bouguettaya & Mistry, 2020*; *Ibrhim et al., 2020*; *Balliu, Merro & Pasqua, 2020*). A modicum of researches focused on providing literature reviews about conflict detection (*Resendes, Carreira & Santos, 2014*; *Carreira, Resendes & Santos, 2014*), but there exist abundantly of studies that proposed classification for conflicts in their work (*Sun et al., 2014*; *Shah et al., 2019*; *Chaki, Bouguettaya & Mistry, 2020*). According to the study of the several literature and approaches on IoT-based apps related conflicts. We have proposed a conflicts' classification that provides a-priori technique for detecting and mitigating intra-user' rules bugs, conflicts, and policy violations. We mean by **a-priori**, the ability to conclude that the system encountered a bug or conflict before its occurrence.

Specifically, we have conducted a qualitative comparison of available approaches based on the proposed conflicts' classification. We focus on two different aspects: coverage of conflict types and inclusion of automation services and system policies. The advantages of our conflicts' classification are: (1) considering the first attempt towards a comprehensive classification of rule-based conflicts in IoT systems, (2) supporting the inclusion of automation services and system policies in conflicts definitions, (3) using Satisfiability Modulo Theories (SMT)-based model-checking (*Barrett & Tinelli, 2018*) for defining the conflicts instead of creating manual annotations, (4) considering a step toward developing a custom compiler/verifier to distinguish bugs and conflicts in IoT-based apps, and (5) highlighting the uncovered conflicts in the literature that should be taken in consideration.

In this survey, we conducted a comprehensive literature review of IoT apps conflicts and introduced a classification of these conflicts. To the best of our knowledge, the proposed classification of conflicts in this paper surpasses previous surveys (*Resendes, Carreira &*

*Santos, 2014*; *Carreira, Resendes & Santos, 2014*; *Sun et al., 2014*; *Shah et al., 2019*; *Chaki, Bouguettaya & Mistry, 2020*) by covering more levels and types of conflicts. Moreover, this study highlights the limitations of the current IoT apps verification frameworks with the help of the proposed classification. Moreover, it proposes a solution to fill the gap in the current IoT systems.

This survey provides an extensive illustration of current technological and research issues of IoT service bugs and conflicts which is vital for IoT systems community. For example, companies that work in developing IoT automation frameworks will use our classification. This is done by developing a new framework that covers all mentioned conflicts to detect end-users programming bugs, ensure maximizing safety and security. Also, companies that work in building smart cities will have a clear vision and evaluation of the current IoT systems with the aid of this survey. On the other side, researchers in the IoT systems verification area will gain benefits from the paper through our recommendations of how all mentioned conflicts can be checked by showing the shortage of using a single model checker. This paper urges researchers of the importance of developing IoT verification systems using hybrid model checkers and approaches as using a single method/model checker would not be enough to cover all conflicts.

This paper is structured as follows. In the 'Survey Methodology' section, the searching and inspecting process is conducted to obtain the most related works to the topic of this paper interest. The 'Current Conflicts' Classifications and Detection Methods' section investigated the current relevant work in detail. In 'The Proposed Conflict' Classification Framework' section, a classification of conflicts' levels and types is proposed with a definition and explanation of each conflict. In the 'Discussion' section, based on a qualitative comparison of relevant studies, we provided some insights and limitations in the current IoT apps conflict detection frameworks that should be considered when developing IoT systems. Finally, the 'Conclusions' section provides a concluding summary.

# SURVEY METHODOLOGY

We conducted a searching process method for the literature review. This process specifies research questions and defines some criteria for including and excluding the studied papers.

## Overarching research question

From the authors' point of view, to develop an IoT framework that guarantees user comfort with an acceptable level of system correctness and safety, conflicts that may occur should be well-defined and known. For this, the analysis of the literature was guided by the following research question:

*How much coverage of conflict types is achieved by the current IoT verification and detection frameworks?*

To answer this question, we need to determine if IoT apps conflict detection is still a relatively challenging task? Also, is it essential to develop a new IoT apps verification framework or not? These questions are compulsory identifying the current state-of-the-art of IoT-based end-users automation conflicts' classifications and detection approaches:

- What are the existing IoT apps conflicts' classifications?

- What are the existing IoT apps' conflicts detection methods?
- What are the main characteristics of each conflict detection method?

## Search process

To ensure that the survey is rigorous and unbiased, we conducted a review of relevant studies that address the research questions. Figure 2 shows the search process used to obtain relevant studies. The search process involves five phases of search and enhancement. The search process performed in well-known online databases, which are IEEEXplore (*IEEE, 2021*), ACM (*ACM, 2020*), Google Scholar (*Scholar, 2020*), Springer (*Springer, 2021*), and ScienceDirect (*ScienceDirect, 2020*).

In phase 1, the search string Eq. (1) below is created to retrieve relevant studies from aforesaid databases. We formulated this search string based on an analysis of the keywords from the relevant literature. Initially, using these search terms, we obtained plenty of potential studies in the databases. Totally, the search string produced 1536 results.

$$(\text{``}\textit{IFTTT''}\text{ OR ``}\textit{ECA rules''}\text{) AND ``}\textit{conflict''}\text{ AND }(\text{``}\textit{detection''}\text{ OR ``}\textit{classification''}\text{)} \tag{1}$$

In phase 2, a step-forward to minimize the results from phase 1, we filter results by refining the search string Eq. (1) by adding more keywords related to IoT systems conflicts as below in search string Eq. (2). As a result of refining the search terms, 223 studies were identified.

$$(\text{``}\textit{IFTTT''}\text{ OR ``}\textit{ECA rules''}\text{) AND ``}\textit{conflict''}\text{ AND (``}\textit{detection''}\text{ OR ``}\textit{classification''}\text{) AND}$$
$$(\text{``}\textit{IoT''}\text{ OR ``}\textit{building automation''}\text{) AND (``}\textit{correctness''}\text{ OR ``}\textit{safety''}\text{)} \tag{2}$$

In phase 3, due to paper unavailability, redundancy, or irrelevance, we excluded those papers that satisfy these reasons. We obtained 150 studies.

In phase 4, we defined inclusion and exclusion criteria to enhance the filtering results from phase 3. Below, the inclusion and exclusion criteria used. We obtained 100 studies.

- Inc1: including the papers that provided conflict classification.
- Inc2: including the papers that described a conflict detection method.
- Exc1: excluding the papers that did not describe a conflict classification or provide a conflict detection method.

In phase 5, after using the filtering search string, inclusion and exclusion criteria, and excluding unavailable and duplicated papers, we ensure that the remaining papers are mostly related to the IoT apps conflict classifications and detection by screening both papers' titles and abstracts. Finally, we obtained 63 studies. Figs. 3 and 4 show the distribution of reviewed papers over the defined period (between 2014 and 2020) and selected search databases, respectively.

## CURRENT CONFLICTS' CLASSIFICATIONS AND DETECTION METHODS

Developing reliable software means to ensure the correctness, security, and safety issues in the to-be software. Among the stumbling blocks that make developing such software

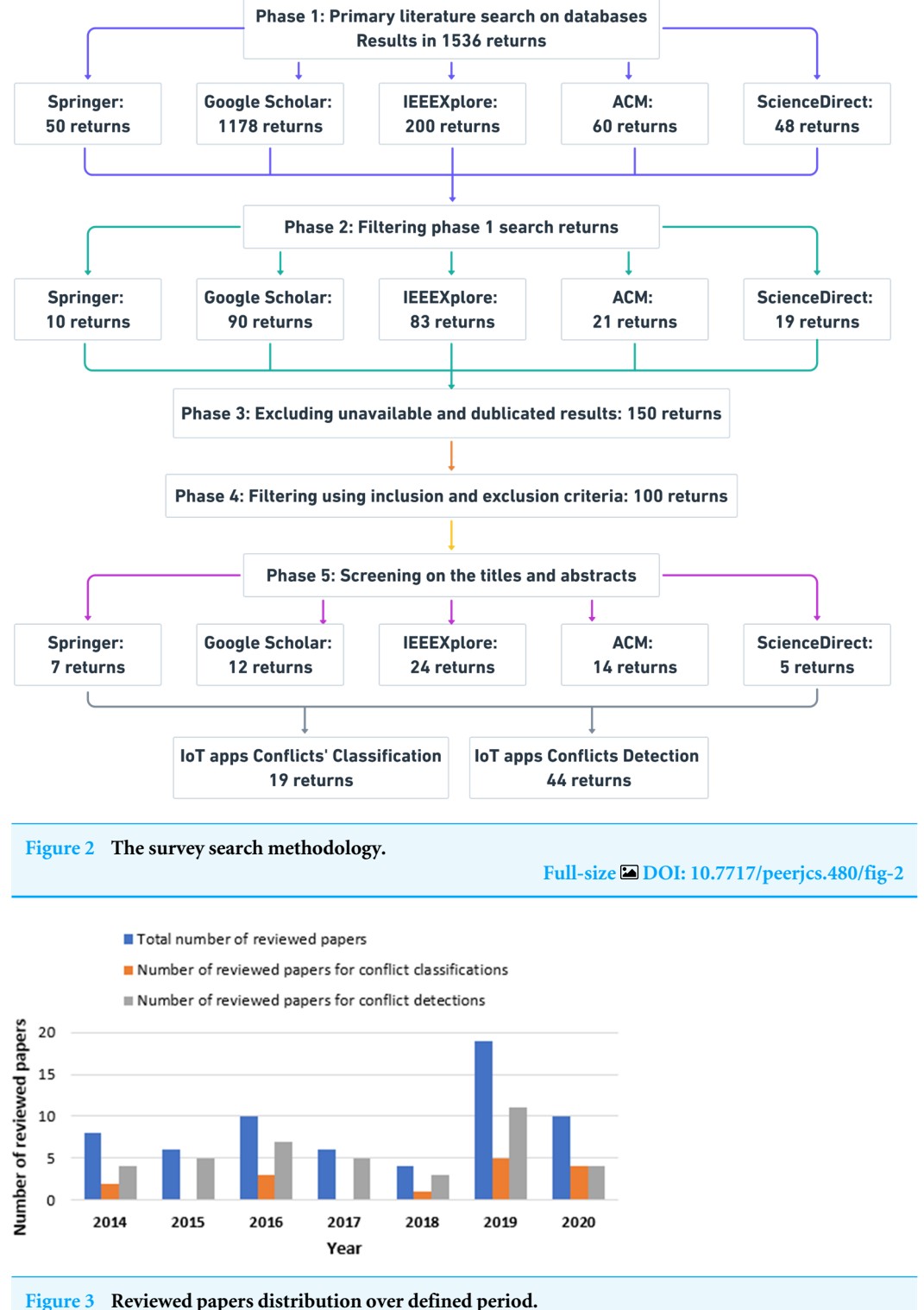

**Figure 2** The survey search methodology.

**Figure 3** Reviewed papers distribution over defined period.

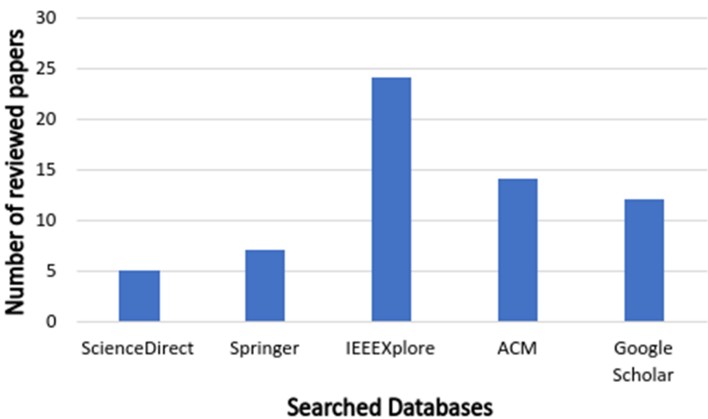

**Figure 4** Reviewed papers distribution over selected search databases.

is still a challenge are the errors related to the software, violating certain properties, and vulnerabilities. Ignoring the surety for these issues could lead to negative impacts either on the software, the users, hardware failures, sniffing users' sensitive information, the harmful influence of information flow, or other ramifications.

A plethora of defense techniques and approaches have been proposed and applied in different domains and contexts for the above-mentioned issues. Among these techniques are **Security Policy Enforcement** (*Herrmann & Murari, 2004*; *Sicari et al., 2016*; *Neisse, Steri & Baldini, 2014*; *Keromytis & Wright, 2000*; *Adi, Hamza & Pene, 2018*), **Language-based Security** (*Abadi, Morrisett & Sabelfeld, 2005*; *Bandi, Fellah & Bondalapati, 2019*; *Khan et al., 2019*; *Vaidya et al., 2019*; *Zigmond et al., 2019*), **Formal-language Verification** (*Foughali et al., 2018*; *Fragoso Santos et al., 2017*; *Grimm, Lettnin & Hübner, 2018*; *Abbas et al., 2020*; *Halima et al., 2018*), **Software Verification** (*Rodriguez, Piattini & Ebert, 2019*; *Feldt et al., 2010*; *Zheng et al., 2014*; *Miksa & Rauber, 2017*; *Zhang et al., 2019b*), and **Developing Secure Compiler** (*Busi & Galletta, 2019*; *Lee, Jeong & Son, 2017*; *Zúñiga et al., 2020*; *Abate et al., 2020*; *Hastings et al., 2019*). All of these methods and others share the same approach. Giving a program as input along a set of predefined requirement properties/policies (e.g., the syntax and semantic behaviors' rules, security/safety needs, software constraints, etc.) that needed to be validated in the input program, and concluding with either the given program is passed (i.e., satisfying all the given requirements) or failed (i.e., some bugs/conflicts are issued due to violating these requirements). The input programs and the predefined requirements are differed according to the context they applied for, how they are represented, the checking/validating process, and how to resolve the violated requirements.

In the IoT context, especially for the automation systems, correctness, security, and safety consider sensitive concerns (*Tawalbeh et al., 2020*) as a misunderstanding or mishandling them may lead to harm to the users (e.g., unlocking doors at the wrong time) or even causing massive damages in IoT-system environment (e.g., not opening water valve in fire situations). Regarding the IoT app's source code, static and dynamic analysis tools (*Celik et al., 2018*; *Celik, McDaniel & Tan, 2018*; *Celik, Tan & McDaniel, 2019*) have been

proposed to track the privacy-sensitive data flow and to prevent conflicts between IoT apps interactions. However, the IoT context, in general, has some differences, which make IoT apps source code errors prediction and mitigation techniques than its peers in the traditional programs analysis. Among these differences there are:

- The (computation, memory, and energy) resource limitations of IoT systems impose different constraints,
- The dynamic nature of IoT systems implies the need for dynamic and fault-tolerant services verification and coordination methods matching this nature,
- The new nature of interaction with the physical world which makes the traditional methods need a bit more refinement or even inventing new methods, and
- The multi-user programming environment used in IoT systems results in a different set of properties to be checked which depends on the end-users preferences.

To a certain extent, the proposed work in this paper is targeting the same goal as the above-mentioned techniques, in which it provides a classification for the programming bugs and safety policies violations in IoT systems. However, there are some similarities between these methods, especially the compiler design, and the proposed work in this paper as follows: (i) each has programmers, but in IoT automation systems the end-users are considering the main programmers instead of developers, (ii) each has programmed apps, but here the automation apps are simple (i.e., IFTTT-style rules) instead of hardcoded instructions (e.g., Java source code), and (iii) like the syntax/semantic analysis related to compilers which is responsible for detecting the syntax structure errors and semantic violations in the program code, the syntax/semantic analysis is applied in customized ways in IoT apps.

In next subsections, we reviewed the relevant works in two categories, including: (1) literature that provided conflicts' classifications/categorizations for the IoT-based apps' interactions and (2) IoT-based apps conflict detection approaches. The striking similitude between various tools and approaches denotes that sometimes it is hard to identify intense boundaries between them. Here, we mainly focus on the tools and models contributing to provide a classification or conflict detection in the IoT systems context.

## Literature analysis of conflicts' classification

A body of current researches focused on providing classifications of the conflicts in the context of IoT smart environment or building automation. In the following paragraphs, we reviewed relevant works on conflict classification.

For Ambient Intelligence (AmI) systems, *Resendes, Carreira & Santos (2014)* and *Carreira, Resendes & Santos (2014)* investigated and provided a conflict classification based on four dimensions, (i) source, (ii) interventions, (iii) time of detection, and (iv) solvability. Their classification provided multi-level categories for conflicts and supports the interactions either in user application, between the user application and space policy (system policy), or between different users' applications. However, their work is limited in conflict classification and does not provide an exact or formal definition for the conflicts,

or how they could be occurred. Also, all conflict categories are for one conflict type that is inconsistency conflict.

*Resendes, Carreira & Santos (2014)* and *Homola et al. (2015)* investigated the importance of conflict resolution for Knowledge Representation (KR) in AmI, where agents (i.e., user automation needs) required/consumed knowledge may cause conflicts in their interactions. *Homola et al. (2015)* mentioned and explained approaches that can help to solve these conflicts such as context modeling, multi-context systems, belief revision, ontology evolution, and debugging argumentation, preferences, and paraconsistent reasoning. However, the work focused on the conflict classification proposed in *Resendes, Carreira & Santos (2014)* and did not consider conflict resulting from violating system policies.

*Homola & Patkos (2014)* followed the previous two works in conflict classification and proposed a new conflict category caused by knowledge sharing between different AmI systems. They also referred to five conflict types of this category, which are (i) sensory input, (ii) context, (iii) domain and background knowledge, (iv) goal, and (v) action conflicts. Their work is ignoring conflicts against policy constraints.

The UTEA (User, Trigger, Environment entity, and Actuator) model proposed in *Sun et al. (2014)* provided a classification for IoT apps conflicts. Their classification contains five conflict types (i) shadow conflict, (ii) execution conflict, (iii) environment mutual conflict, (iv) direct dependence conflict, and (v) indirect dependence conflict. This classification is based on 11 rule relations. Although they provided how the conflict occurs, *Sun et al.* considered only the interaction between rules and the conflicts when users' priorities are similar. Their rule relation schema depended on manual annotations of conflicts and no representation for environment and system requirements.

Based on Feature Interaction (FI), *Magill & Blum (2016)* provided five types of rule interaction and defined them as conflicts. These interactions are (i) shared trigger interaction, (ii) sequential action interaction, (iii) looping interaction, (iv) multiple action interaction, and (v) missed trigger interaction. Using Event Calculus (EC) with proposed detection algorithms, authors capable of detecting these conflicts. However, the work did not explain how these conflicts may appear in the run-time of automated service.

For WSAN-based smart building, *Sun et al. (2016)* provided a conflict classification and categorized them into two based (i) rule conflict on the device (redundancy and contradiction conflicts) and (ii) rule conflict on the environment (write-write and read-write conflicts). The proposed algorithms in their work missing the consideration of system policies and consequently forgetting its related violations.

IoTSAT framework is proposed in *Mohsin et al. (2016)* to detect the security risks in IoT devices. The framework is based on SMT solvers to convert each of the functional and network-level dependencies to predicate logic. For ensuring the behavioral model of IoT, some constraints are defined. The threats are classified into three types (i) context threats, (ii) trigger threats, and (iii) actuation threats.

While, HomeGuard system proposed in *Chi et al. (2018)* for ensuring smart home safety against threats resulting from interactions between multiple IoT apps. A categorization of the Cross-App Interference (CAI) threats has been proposed, and the detection of these threats was based on a symbolic detection module. The categorization includes

three categories (i) action-interference threats (actuator race and goal conflicts), (ii) trigger-interference threats (covert triggering, self disabling, and loop triggering threats), and (iii) condition-interference threats (enabling-condition and disabling-condition threats). Although the authors cover a range of threats between IoT apps, they omit the representation of a system or environment constraints and the threats that may occur due to violating these constraints.

For security issues, the iRULER tool in *Wang et al. (2019)* is proposed to discover inter-rule vulnerabilities. These vulnerabilities are (i) condition bypass, (ii) condition block, (iii) action revert, (iv) action conflict, (v) action loop, and (vi) action duplicate. iRULER is based on natural language processing (NLP) methods to derive flows in trigger-action apps and verify the inter-rule interactions using Satisfiability Modulo Theories (SMT) (*Barrett & Tinelli, 2018*) based model checking. However, the tool not supporting complex rules (rule conditions are represented only using Boolean flags) and does not take into consideration the conflicts resulting from violating system policies.

An error categorization is explained in *Palekar, Fernandes & Roesner (2019)* for user programming errors in smart home behaviors. They mentioned errors like (i) lack of action reversal, (ii) feature interaction, (iii) feature chaining, (iv) event+event rules, (v) state+state rules, (vi) missing trigger, (vii) missing action, (viii) secrecy violations, and (ix) integrity violations. Their work mentioned the missing of some trigger-action platforms in handling these errors. However, the work missing to mention the level of conflicts, they only focused on checking these conflicts during the user authoring process.

Using TAP rules, *Brackenbury et al. (2019)* explained using testing the bugs related to the rules. They categorize bugs into three categories, (i) control flow, (ii) timing, and (iii) inaccurate user expectations. Although the work provided more conflict types, they did not make allowances for the bugs that may result from policy violations and composite service violations against policies. *Shah et al. (2019)* proposed a schema to detect rule conflicts, which include execution, independent, shadow, and rule incompleteness conflicts. But, considered only conflicts in actions of the rules (e.g., contradiction or false impact actions) and no representation for environment or system requirements.

A conflict taxonomy and detection model are proposed in *Alharithi (2019)*. Conflicts are considered for multiple user-defined policies' interactions. The taxonomy categorizes the conflicts into two main categories, (i) direct conflicts (e.g., opposite conflict, overwrite conflict, and environmental conflict) and (ii) indirect conflicts (e.g., chain conflict, feedback conflict, and environmental conflict). Static analysis is used to detect the interaction between IFTTT rules. Although the taxonomy provided a range of policy conflicts, it did not provide methods to resolve the detected conflicts. Also, it did not take into consideration the conflicts that may result due to violating the system policies.

The work presented in *Huang, Bouguettaya & Mistry (2020)* investigated the rule interactions in single user's rules by proposing an ontology to represent different contexts of IoT applications. Using this ontology, the framework categorized the conflicts into three categories (i) environment related conflicts (i.e., opposite environment conflict, additive environment conflict, and transitive environment conflict), (ii) action related conflict (i.e., contradiction conflict), and (ii) quality related conflict (i.e., contradiction conflict only for

Integer devices). However, detecting conflicts is based on knowledge-based IoT-services (conflict annotations) which written by a domain expert. Also, based on the setting of time threshold, which requires techniques to determine the suitable ones. Moreover, the same IoT service property is used for two different conflicts, which may confuse the detecting process (e.g., brightness is used for quality conflict and additive environment conflict).

In *Chaki, Bouguettaya & Mistry (2020)* and *Chaki & Bouguettaya (2020)* a hybrid framework to detect conflicts is proposed. The framework has a knowledge-driven approach represented in an ontology to model and describe the conflicts in both functional (i.e., On/Off of a device) and non-functional (e.g., Temperature value) properties of IoT services. Also, it has a data-driven approach represented using IoT service usage history. However, conflict detection is based on peoples' service usage history, which not suitable for real-time (dynamic) IoT service requests. Policy violations are neglected.

*Trimananda et al. (2020)* provided a study of pairwise IoT apps interactions to determine app conflicts. They categorize the conflicts into three groups, (i) conflicts resulting from accessing the same device with incompatible values, (ii) conflicts resulting from physical interactions as app chain, and (iii) conflicts resulting from modifying the same global variable. To detect these conflicts, the IoTCheck tool is implemented based on the Java Pathfinder (JPF) model checking infrastructure (*Visser et al., 2003*). A drawback of IoTCheck is that, the IoTCheck's performance due to the use of JPF and Groovy language (*Groovy, 2020*). Moreover, IoTCheck focused only on three types of conflicts ignoring conflicts between multiple app interactions and system policy violations.

The IoTCOM approach is proposed in *Alhanahnah, Stevens & Bagheri (2020)*. IoTCOM is based on the static analysis of behavioral models of IoT apps and formal methods to detect the security and safety violations against some predefined IoT safety and security properties. IoTCOM classifies the potential multi-IoT app coordination threats into seven classes, (i) action-trigger, (ii) action-condition (match), (iii) action-condition (no match), (iv) self coordination, (v) action-action (conflict), (vi) action-action (repeat), and vii) exclusive event coordination. Although IoTCOM investigates the coordination between IoT apps, it is omitting conflicts resulting from properties violation and co-exists of IoT apps that violate these properties. Also, IoTCOM did not provide resolution methods for the detected threats.

## Literature analysis of conflict detection

Different methods and approaches are developed for ensuring the interactions between automation services and system policies in IoT systems by detecting various conflicts that maybe happened in these interactions. The following is a brief explanation of current IoT app conflict detection methods.

### Single User IoT App Conflicts Detection

In *Cano, Delaval & Rutten (2014)*, the authors categorized the ECA rules problems based on compiler and runtime. The ECA rules are translated formally to Heptagon/BZR language. Rules problems like (i) redundancies, (ii) inconsistencies, (iii) circularities, and iv) app dependent safety issues are detected. Termination, confluence, and consistency

are explained in *Corradini et al. (2015)*. The ECA rules are created using IRON language and compiler proposed in their work. However, the proposed IRON language required the programmer to have knowledge about devices' properties (e.g., device type, physical or logical—has a constraint or not) which implies a burden for normal users when developing their apps. Also, it did not provide a means for the conflicts caused by violating system-specific constraints.

Inconsistency and desired behavior violation conflicts in IFTTT-style programs are investigated in *Huang & Cakmak (2015)* using some mental models. The conducted mental models studies concluded with the ambiguities cause errors in programs. Conflicts against system requirements are omitted in their work. Temporal ECA rules are used by *Le Guilly et al. (2016)* and provided a framework to define behaviors that are not wanted to occur in the environment. Their work defined a set of constraints represented in the undesired behaviors states. Rules used in their work are simple (a single condition and action) and did not support complex rule systems. The IoT apps are formed using a temporal ECA language and checked against the reachability of these constraints. However, only the reachability is checked and not touching conflicts and violations that may take place.

*Perumal et al. (2016)*, action contradiction conflict was discussed. Also, an ECA Priority Schema has been proposed to solve this conflict. However, the work did not take into account other conflict types and no representation for system policies. Policy violation like condition inconsistency and non-specified conflicts that occur in IoT apps has been studied by *Liang et al. (2016a)* using Salus framework. However, the work focusing only on two conflicts, ignoring other conflicts and local conflicts.

*Peña et al. (2016)* proposed a decision support system using data mining techniques and context-aware information to detect energy inefficiency situations that cause high energy consumption in smart buildings. The detection is based on a developed set of energy efficiency indicators to find these energy anomalies. But, the energy efficiency indicators policies are considered rigid and not suitable for the dynamic nature of IoT applications.

*Vannucchi et al. (2017)* conflicts such as Unused, Redundant, and Incorrect rule conflicts that may happen in Integrated Rule ON data (IRON) rules are referred. These conflicts have been detected during different rule interactions, so no handling for local conflicts. *Goynugur et al. (2017)* authors proposed a policy (i.e., user-defined rules) framework based on OWL-QL representation language. They also used the JSHOP2 AI planner to avoid conflicts. In their work, two policies cause a conflict if and only if: (a) policies shared the same device or individual, (b) one policy must oblige an action, while the other prohibits the same action; and (c) policies are active at the same time. Their conditions for conflict are considered only for direct conflict and not for other conflicts that may occur in design and run times.

In *De Russis & Monge Roffarello (2018)*, a debugging approach for trigger-action rules against loops, action inconsistency, and redundancy conflicts have been proposed, the proposed approach—also in *Corno, De Russis & Monge Roffarello (2019)*—was based on semantic colored Petri net to formulate the execution model of rules. For ensuring not violating end-users privacy, a static taint analysis tool (SAINT) for IoT applications is proposed in *Celik et al. (2018)*. SAINT is based on analyzing the Samsung SmartThings

app's source code and data packets to prevent and track any attacker malicious interactions. SAINT privacy-sensitive data is divided into device states, device information, location, user inputs, and state variables.

My IoT Puzzle tool has been proposed in _Corno, De Russis & Roffarello (2019)_ to compose and debug IFTTT rules. The rules are authored using the block programming approach to achieve simplicity for users. The authors define three types of conflicts' loops, redundancies, and inconsistencies. However, the tool correctly detects the rules problems, but the mentioned conflicts are for design-time only. In this work (_Al Farooq et al. 2019_), a rule-based system called $IoTC^2$ has been implemented in Prolog to ensure safety in controllers' behaviors through some controller safety policies defined by _Al Farooq et al. (2019)_ and conflicts such as dependent/indirect dependent and overlapping events. However, the rigidity safety properties defined in $IoTC^2$ restrained the dynamic nature of IoT applications.

_Manca, Santoro & Corcella (2019)_ explained how end-user trigger-action rules could be analyzed and debugged through why/why not common questions to ensure it behaves as it desired by the user. The Interactive Trigger-Action Debugging (ITAD) tool developed in _Manca, Santoro & Corcella (2019)_ is capable of detecting some conflicts between user-defined rules. Although ITAD refers to there are design and runtime conflicts, it omits to represent system policies. AutoTap has been proposed in _Zhang et al. (2019a)_ to allow less user program failure through ensuring means safety-properties which allow users to express system states that should always be satisfied. The user rules are translated to LTL for synthesizing against these properties. AutoTap is missing other conflicts that may occur due to sharing the location of interest between different users, also omitting local conflicts.

_Krishna et al. (2020)_ proposed a tool called MOZART (Mozilla and Advanced Rule Triggers) that provides several functionalities for IoT applications, such as providing UI to create rules, analyzing the created rules and checking against deadlocks, and finally deploying to execute the rules. MOZART only checks for deadlock and did not care about errors resulting from devices' effects on each other. Deadlocks are checked after forming the composite service using the suggested operators.

### Multiple user IoT apps conflicts detection

DepSys (_Munir & Stankovic 2014_) focused on dependency checks that may take place between different IoT apps that share devices in both service times (design and run times). DepSys defined some dependencies requirements to ensure the IoT apps relationships. But, the developers are responsible for specifying these dependencies within their apps' metadata. Although DepSys provided different types of app dependencies, it did not consider other conflicts that happened in runtime when violating system constraints.

Based on a multi-agent auction mechanism, _Liang, Hsu & Lin (2014)_ proposed different algorithms to efficiently orchestrate IoT applications that share actuators. The authors focused on action contradiction conflict between two applications. However, the work benefits from a multi-round single-item auction in conflict resolution to maximize the utility, but they did not take into account other conflict types.

*Cheng et al. (2014)* proposed an IoT middleware for conflict resolution. Concurrent access for devices/resources is the main conflict focused in the middleware where situation-aware services are represented as Situation-oriented ECA rules. Based on the diagram of the situation state transition, the author proposed a set of policies to resolve the conflict between users. The middleware ignored other types of conflicts.

Safe Internet oF Things (SIFT) (*Liang et al., 2015*) automatically verified IoT apps correctness against high-level user requirements (policies) using an SMT-based checker. The correctness in their work represented the avoidance of simultaneous device access and detecting the violations of predefined policies. SIFT ignores other conflicts of policy violation and co-exist of multiple users.

In *Yagita, Ishikawa & Honiden (2015)*, the authors presented an approach to handle a single actuator simultaneously control conflict at installation time. The rules generated by the parser module in *Yagita, Ishikawa & Honiden (2015)* are translated to a Kripke structure and verify their correctness against one property, "*no two apps use actuators to create different effects at the same location*", through the use of model-checker. Also, using the effect attribute of actuators, the approach can detect device-influence conflicts. Different other conflicts are ignored in this approach.

*Zave, Cheung & Yarosh (2015)* proposed a framework for controlling, composing, and conflict resolution, IoT apps. IoT apps are represented using feature interactions to check their behavioral properties using the Alloy Analyzer. The only conflict reported in their work is the inconsistency conflict. But no clear definition for other conflicts.

An Agent-based Negotiation System (ABNS) has been proposed in *Alfakeeh & Al-Bayatti (2016)* to simplify interactions between different residents' services expressed using means of features interaction hierarchy. Contradiction conflict caused by concurrent access for shared devices is discussed. No other conflicts are detected in their work.

TrigGen tool is proposed in *Nandi & Ernst (2016)* for finding the missing conditions or preventing unnecessary triggers based on static analysis of OpenHAB rules. It depends on generating the abstract syntax tree (AST) of the rule actions to suggest adding or eliminating triggers. TrigGen rules are depending only on OpenHAB rules which may not be suitable to apply on other platforms. And, saying the rule is missing or requiring triggers is based on violating the user's expectation, which can not be obtained previously and requires effort-intensive to be detected.

*Hadj et al. (2016)* added a modification to iCASA (*Escoffier, Chollet & Lalanda, 2014*) by adding an autonomic access layer, which allows detecting direct conflict between smart home apps due to shared devices. The work not suitable for non-technical end-users and also did not provide management for other conflicts.

A context descriptor using a visualization model proposed in *Oh et al. (2017)* for trigger contradictory action conflict that may happen between apps and In-deterministic conflict occurs against system policies. No clear definition of the mentioned conflict. Simultaneously arrived conflict for events has been explored in *Shahi et al. (2017)* and depending on weighted-priority scheduling for solving such conflict. The work focused on one conflict, ignoring other conflict types.

Based on Trigger-Actuator-Status (TAS) rules, *Lin et al. (2017)* proposed a formal rule model for just one conflict, redundancy conflict. However, the rule model did not consider other conflict types.

In *Celik et al. (2019)* the authors provided two frameworks Soteria (*Celik, McDaniel & Tan 2018*) and IoTGuard (*Celik, Tan & McDaniel 2019*) for real-world IoT apps verification. The former performing static analysis against identified properties. The latter focused on ensuring safety and security proprieties during runtime IoT apps interactions. Both frameworks are based on model checking (e.g., NuSMV *Cimatti et al. (2002)*). For IoTGuard, to detect policy conflict using the unified dynamic model of apps interaction, these apps must have a shared events chain between them, so it fails to detect conflict in IoT apps (e.g., Joint-behavior services conflict in Subsection 'Conflict Notations and Types') not having this events chain.

OKAPI platform is proposed in *Melissaris, Shaw & Martonosi (2019)* for eliminating the consistency deficiencies, event reordering, and race conditions problems resulting from accessing or modifying shared resources in smart homes. The consistency check did not include the system requirements violations. RemedIoT framework proposed in *Liu et al. (2019)* to detect and resolve IoT app conflicts with respect to policies—set of rules that must not be violated—through simplifying IoT app using the Abstraction Module. There are three types of conflicts defined in this work, Racing events (i.e., contradicting actions), Cyclic events, and policy violation conflicts. The work did not take into account other different conflicts in the app itself or between apps.

To simplify IoT apps abstractions and support free-conflicts apps, VISCR proposed in *Nagendra et al. (2019)* and *Nagendra et al. (2020)*. The IoT infrastructure administrators' policies and the communication between devices policies are translated into a vendor-independent graph-based specification to detect some conflicts like loops and runtime violations and rouge policies. VISCR can detect a small set of conflicts and bugs. VISCR missing of representing environment policies which have a great impact to secure and safe occupants and building.

In *Li, Zhang & Shen (2019)* and *Shen, Zhang & Li (2017)* utilizing graph-based representation and SMT solvers to represent IoT inter-app interplays and detect conflicts between IoT apps. Using Inter-App Graphs (IA-Graph) and some proposed algorithm provided the Detector of IoT App Conflicts (DIAC) tool for checking the conflicts. Also, they provided a definition refinement for the apps conflicts based on three categories (i) Strong conflict, (ii) Weak conflict, and (iii) Implicit conflict. These conflict categories investigate the inter-app influences and the action inconsistency conflict between them. Although DIAC using an SMT solver, there are some conflicts ignored or can not be detected due to missing the relationship analysis between IoT apps rules.

*Xiao et al. (2019)* proposed a method called Automatic and Interpretable Implicit Interference Detection (A3ID) to detect implicit interference based on intensive analysis using natural language processing (NLP) techniques and a lexical database. But, the detection process is time consuming due to intensive rule NLP analysis, also based on the knowledge extracted from the knowledge graph, which still has a weakness of representing all information of all devices.

*Chen et al. (2019)* proposed IoT Interaction Extraction (IoTIE) tool to explore the cross-interactions between IoT Apps and the physical environment entities (e.g., temperature, humidity, light, motion). IoTIE is based on the NLP method to detect these cross-interactions by calculating the similarity between the rules and physical environment entities. However, depending on NLP methods may result in incorrect similarity results, which affect the resolution of these interactions. Also, IoTIE did not care about other types of conflicts that may occur between IoT apps.

Using wireless communications, (*Gu et al., 2020*) introduced the "wireless context" concept that represented the actual packets workflow of IoT apps. By applying the machine learning model, system anomalies (e.g., App misbehavior, Event spoofing, Over-privilege, Device failure, and Hidden vulnerabilities) can be detected by comparing the user IoT context against the wireless context. For detecting hidden vulnerabilities in interactions between apps, hidden channels (i.e., device influences) is used. Also, no solutions for the detected anomalies and no means for representing environment constraints.

*Balliu, Merro & Pasqua (2020)* and *Balliu, Merro & Pasqua (2019)* proposed a framework to ensure the security and safety for cross-app IoT interactions. Based on the process calculus, the authors formally defined safe and secure cross-app interactions policies. Although the proposed policies can make inferences for both syntactic and semantic conditions for IoT apps, they did not provide ways to resolve these interactions. Also, they did not take environmental requirements into considerations when two systems of apps are interacting together.

Rule Verification Framework (RVF) was proposed in *Ibrhim et al. (2020)*. RVF comprised a set of algorithms for detecting and resolving building automation system-related conflicts. RVF is based on Z3 (*De Moura & Bjørner, 2008*), a state-of-the-art SMT-based model checker, to analyze, detect, and resolve the conflicts in campus buildings context. However, RVF fails to detect the co-existence of users' services that violating system policy. Also, no means for representing devices' influences.

To sum up, there are different methods and approaches that have been proposed to categorize and define IoT apps' interaction errors and conflicts. There is a general point of missing in these approaches that is, the conflicts have a rapport with these interactions ignoring either to represent the environment requirements or to check their violations. Also, the detection approaches focus on inconsistency, redundancy, and dependency conflicts. The proposed conflicts classification aims at filling this gap.

## THE PROPOSED CONFLICTS' CLASSIFICATION FRAMEWORK

In this section, we provide a detailed explanation of the proposed IoT-based service conflicts' classification. We begin with the formalization for both automation services and policies defined in the 'Introduction' section; then, we provide the definitions and notations for the conflicts included in the classification.

## Rule and constraint formalization

The rule and the constraint forms for representing user' automation service or system policy used in this work are as follows:

**Simple trigger** A rule structure or policy constraint that specifies a Boolean expression over one or two devices.

```
Rule1: IF Occupancy = True THEN Light = True
Rule2: IF Temperature>20 AND Occupancy = True
THEN Fan = True AND Window = False
```

**Arithmetic operation** A rule structure for specifying a Boolean expression that involving an arithmetic operation in its condition part.

```
Rule1: IF OutTemperature - InTemperature >= 10 THEN AC = True
AND Thermostat = 22
```

**Admissible range** A rule condition part or a policy constraint that identifies a specific range of valid values for a device.

```
Rule1: IF 20 < Temperature < 27 THEN Window = True
Constraint1: 20 < ACThermostat < 27
```

**Device grouping** A policy constraint structure used to specify conjunction or disjunction over a set of devices.

```
Constraint1: AirConditional = True AND Datashow = True
Constraint2: (NOT (AirConditional = True AND Heater = True))
```

**Direct actuation** A rule structure used to specify conjunction over a set of devices that affect system state directly without specifying a condition part.

```
Rule1: Fan = True AND Window = True
```

Translating these rules or constraints requires caring about computer-spoken logic. One of the logic formalisms that may be used to formalize real-life IoT applications is First-order logic (FOL). FOL is used to express IoT applications as an arrangement of action effects on IoT resources.

Among SMT-based languages that formulates the inputs to SMT-based model checkers (e.g., Z3 (*De Moura & Bjørner, 2008*), MathSAT5 (*Cimatti et al., 2013*), Yices (*Dutertre & De Moura, 2006*), and Boolector (*Brummayer & Biere, 2009*)) is the standard SMT-Lib v2 language (*Barrett, Stump & Tinelli, 2010*). Both services' rules and policies' constraints are translated to assertions in the form $C => A$, where $C$ and $A$ are FOL formulae over one or more theories. These assertion in this form means if-this-then-that, or conditional expression that changes the system state as specified in $A$ when the system has a situation state satisfying $C$. The translation using *Barrett, Stump & Tinelli (2010)* will take this general form,

```
(define-fun < variable >  ()  < sort > )
(assert (=>  < assertion > ) ) )
```

where **variable** represents the device name, **sort** represents Boolean or Integer types for devices, and **assertion** represents the transformation of the rule forms (mentioned in Section 'Introduction') as specified by the user.

For instance, the automation service of a lecturer who wants to adjust the indoor environment temperature by letting fresh air entering the lecture hall when the number of students is more than 60. Additionally, suppose a faculty admin who wants to save energy in the lecture hall, saying that, if no one is in the lecture hall, then the air-conditioner should always be off. These examples can be converted to SMT-Lib v2, respectively as follows,

```
(define-fun LectureHallStudentNumber () Int)
(define-fun LectureHallFan () Bool)
(define-fun LectureHallWindow () Bool)
(assert (=> (> LectureHallStudentNumber 60)
          (and LectureHallFan LectureHallWindow)  )  )

(define-fun LectureHallAC () Bool)
(define-fun LectureHallOccupancy () Bool)
(assert (and (not LectureHallAC)
          (not LectureHallOccupancy  )  )   )
```

## Conflict notations and types

*Tuttlies, Schiele & Becker (2007)* specified the conflict concerning end-user service or system policy as "*…a context change that leads to a state of the environment which is considered inadmissible by the application or user*". In our proposed conflicts' classification, as shown in Fig. 5, we identified two main conflict categories. Firstly, **local conflicts** are those conflicts that occur between intra-rules or intra-constraints of the same automation service or policy authored by the same user or the same admin, respectively. The reasons for this type of conflict are the missing of programming erudite of end-users and less knowledge about devices' interaction relationships. Secondly, **global conflicts** are those conflicts that may occur in inter-interactions between different automation services or between different policies. Overlapping in spatio-temporal attributes and violating system policies are the main reasons for these types of conflicts.

These two main categories (local and dynamic conflicts) are related to a level-based category, which indicates where the conflict may occur. There are two levels, **Service Level Conflicts (SLCs)** and **Policy Level Conflicts (PLCs)**. SLCs are the conflicts that may happen in service rules, either local or global. On the other hand, PLCs are the conflicts that may happen in policy constraints, either locally or globally.

For defining conflicts in a low level of a classification hierarchy, Table 1 provides the notations used for describing these conflicts in a formal specification. Also, we started by defining the concept of **Rule Satisfiability Measure (RSM)**. RSM allows us to assess and filter the rules. RSM is the state of the SMT-based checker (i.e., SAT or UNSAT) when checking conjunctions of rule parts (i.e., conditions and actions). In respect of simplifying

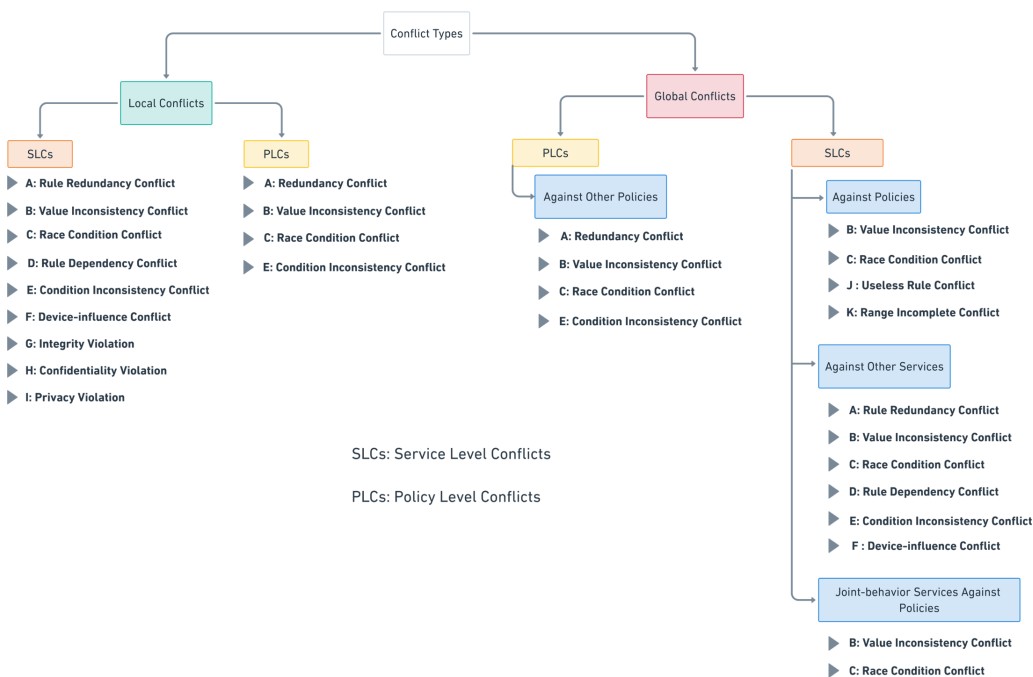

**Figure 5** The proposed IoT apps conflicts' classification.

the conflicts definition, we will use Venn diagram to represent the relation between the rule parts. There are two cases for the diagram. The former case is when the Venn diagram has no superimposition, that represents the ''UNSAT'' state determined by the solver. The latter is when the Venn diagram has a superimposition, that represents the ''SAT'' state determined by the solver.

One or more of the following conflict situations may occur between the two rules under the assumption which they shared the same location and have time overlapping. These conflicts should be taken into consideration to ensure overall system correctness, security, and safety issues, as violating any of them may lead to danger or harm situations.

**Rule Dependency Conflict** (**RDC**) *For two or more rules, this conflict occurs when the actions resulting from satisfying the conditions of one rule leads to trigger the second rule, where its actions lead to the triggering of the first rule, creating an infinite cycle of these rule interactions.*

This conflict is mentioned in *Ibrhim et al. (2020)* and also known as Confluence Property in *Corradini et al. (2015)* and Looping Interaction in *Magill & Blum (2016)*. Figure 6 shows a Venn diagram for this conflict. Formally, this conflict can be described as follows:

$$
\overset{RDC}{\perp} := \begin{cases} \exists R^{ij} \text{ and } R^{ik} \in \text{ the same service i where } j \neq k, \\ \exists d1 \in \{D^{R_c^{ij}}, D^{R_a^{ik}}\}, \ \exists d2 \in \{D^{R_a^{ij}}, D^{R_c^{ik}}\}, and \\ \exists v \in V^{R_c^{ij}} : \ (R_c^{ij} \cap \neg R_a^{ik}) \mapsto UNSAT \ and \\ \qquad\qquad (R_a^{ij} \cap \neg R_c^{ik}) \mapsto UNSAT \end{cases}
\tag{3}
$$

| Table 1 | Conflict notations used. |
|---------|---------------------------|
| **Notation** | **Meaning** |
| *situation* | The state of the system environment at a given location of interest and time and represented by the values sensed by devices attached to the system. It has two cases $situation_v$ and $situation_{iv}$ represent valid and invalid situations, respectively. |
| $V$ | Defines the set of all situations occur in the system. |
| $D$ | Defines the set of all devices in the system. |
| $R^{ij}$ | Represents the $j$th rule which belongs to the $i$th automation service. |
| $R_c^{ij}$ and $R_a^{ij}$ | Represents the conditions and actions groups of the rule $R^{ij}$, respectively. |
| $C^{mn}$ | Represents the $n$th constraint which belongs to the $m$th system policy. |
| $C_c^{mn}$ and $C_a^{mn}$ | Represents the conditions and actions groups of the constraint $C^{mn}$, respectively. |
| $V^{R^{ij}}$ or $V^{C^{mn}}$ | Defines the set of the admissible situations that satisfying the rule $R^{ij}$ or the constraint $C^{mn}$, respectively. |
| $D^{R^{ij}}$ or $D^{C^{mn}}$ | Defines the set of the devices that appear in the rule $R^{ij}$ or the constraint $C^{mn}$, respectively. |
| conflict $\perp$ | Represents the type of the conflict that occurs. |

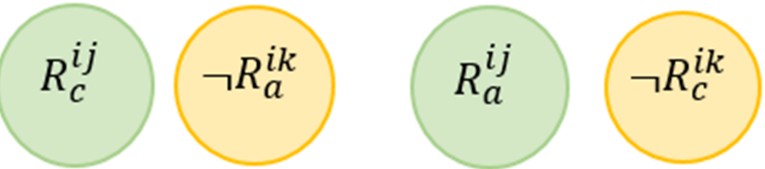

**Figure 6** Rule dependency conflict conflict representations.

```
R11: IF Occupancy = True THEN Light = True
R12: IF Light = True THEN Occupancy = True AND Fan = True
```

Consider the above example, we say that $R^{11} \overset{RDC}{\perp} R^{12}$ according to the definition provided in Eq. (3). In other words, when the rule $R^{11}$ is triggered by a system situation and its actions affects the environment, the second rule $R^{12}$ will be activated, which in turn leads to satisfying the conditions of $R^{11}$, and so on.

**Rule Redundancy Conflict** (RRC) *For any two rules, this conflict may occur when the system has at least one valid situation which satisfies the conditions of the rules and the actions of one rule is a subset of the actions of the second rule.*

This conflict is mentioned in *Ibrhim et al. (2020)* and also known as Shadow Conflict in *Le Guilly et al. (2016)*. Figure 7 shows the Venn diagram for the conflict. Formally, this

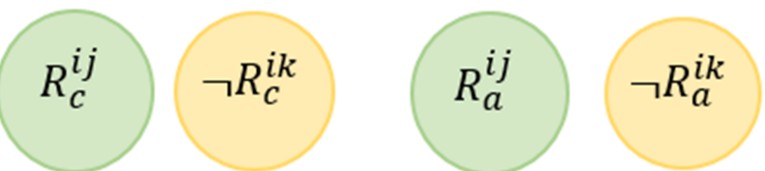

**Figure 7  Rule redundancy conflict venn diagram representation.**

conflict can be described as follows:

$$
\mathop{\perp}^{\text{RRC}} := \begin{cases} \exists R^{ij} \text{ and } R^{ik} \in \text{ the same service i where } j \neq k, \\ \exists d1 \in \{D^{R_c^{ij}}, D^{R_c^{ik}}\}, \exists d2 \in \{D^{R_a^{ij}}, D^{R_a^{ik}}\}, and \\ \exists v \in V^{R_c^{ij}} : (R_c^{ij} \cap \neg R_c^{ik}) \mapsto UNSAT \text{ and} \\ \qquad\qquad (R_a^{ij} \cap \neg R_a^{ik}) \mapsto UNSAT \end{cases} \tag{4}
$$

```
R11: IF Temperature >= 25 AND Occupancy = True THEN Fan = True
AND Window = True
R12: IF Occupancy = True THEN Fan = True
```

Giving the above example, we say that $R^{12} \mathop{\perp}^{\text{RRC}} R^{11}$ (i.e., $R^{12}$ is redundant based on $R^{11}$) according to the definition provided in Eq. (4). In other words, it is not possible to assign values to the variables (Temperature and Occupancy) to satisfy $(Temperature >= 25 \cap Occupancy = True) \cap \neg(Occupancy = True)$. Also, there is no assignment to the variables (Fan and Window) that satisfied this conjunction $(Fan = True \cap Window = True) \cap \neg(Fan = True)$.

**Value Inconsistency Conflict (VIC)** *For any two rules, this conflict may occur when the system has at least one valid situation that leads to the triggering of these rules and their actions change the state of the shared devices in a contradicting way. A service rule and a policy constraint may have this conflict as well.*

This conflict is mentioned in *Ibrhim et al. (2020)* also known as Contradiction Conflict in *Sun et al. (2016)*, Execution Conflict in *Sun et al. (2014)* and referred to as Termination Property in *Corradini et al. (2015)*. Figure 8 shows the Venn diagram for the conflict. Formally, this conflict can be described as follows:

$$
\mathop{\perp}^{\text{VIC}} := \begin{cases} \exists R^{ij} \text{ and } R^{ik} \in \text{ the same service i where } j \neq k, \\ \exists d1 \in \{D^{R_c^{ij}}, D^{R_c^{ik}}\}, \exists d2 \in \{D^{R_a^{ij}}, D^{R_a^{ik}}\}, and \\ \exists v \in V^{R_c^{ij}} : (R_c^{ij} \cap \neg R_c^{ik}) \mapsto UNSAT \text{ and} \\ \qquad\qquad (R_a^{ij} \cap R_a^{ik}) \mapsto UNSAT \end{cases} \tag{5}
$$

```
R11: IF Temperature >= 25 AND Occupancy = True THEN Fan = True
R12: IF Occupancy = True THEN Fan = False
```

Giving the above example, we say that $R^{11} \mathop{\perp}^{\text{VIC}} R^{12}$ according to the definition provided in Eq. (5). In other words, every time $R^{11}$ is triggered, $R^{12}$ is triggered as well, but the actions of

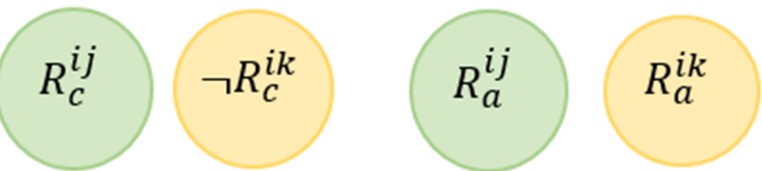

**Figure 8** Value inconsistency conflict venn diagram representation.

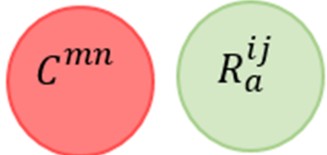

**Figure 9** Value inconsistency conflict against policy constraint venn diagram representation.

the two rules cannot be simultaneously satisfied, as there is no valid assignment that satisfies $(Temperature >= 25 \cap Occupancy = True) \cap \neg(Occupancy = True)$, and there is no valid assignment that satisfies the rules' actions simultaneously $(Fan = True) \cap (Fan = False)$.

$$
\overset{VIC}{\bot} := \begin{cases} \exists C^{mn} \text{ and } R^{ij}, \\ \exists d \in \{D^{C^{mn}}, D^{R_a^{ij}}\}, and \\ \forall v \in V^{C^{mn}} : (C^{mn} \cap R_a^{ij}) \mapsto UNSAT \end{cases} \tag{6}
$$

When VIC occurs between an automation service and a system policy constraint, as shown in Fig. 9, we say that $R^{11} \overset{VIC}{\bot} C^{11}$ according to the second description of $\overset{VIC}{\bot}$ provided in Eq. (6). For example, consider the following service rule and policy constraint. There is no satisfying assignment for $(ACThermostat > 22) \cap (AirConditioner = True \cap ACThermostat = 18)$.

C11: ACThermostat > 22
R11: IF Temperature > 20 THEN AirConditioner = True AND ACThermostat = 18

**Condition Inconsistency Conflict (CIC)** *For two rules, we say that there is a CIC conflict/warning when rules' conditions are contradicting and the actions of one rule are considered a subset of the other rule's actions.*

This conflict is mentioned in *Ibrhim et al. (2020)*. Figure 10 shows the Venn diagram for the conflict. Formally, this conflict can be described as follows:

$$
\overset{CIC}{\bot} := \begin{cases} \exists R^{ij} \text{ and } R^{ik} \in \text{ the same service i where } j \neq k, \\ \exists d1 \in \{D^{R_c^{ij}}, D^{R_c^{ik}}\}, \exists d2 \in \{D^{R_a^{ij}}, D^{R_a^{ik}}\}, and \\ \forall v \in V : (R_c^{ij} \cap R_c^{ik}) \mapsto UNSAT \text{ and} \\ \qquad\qquad (R_a^{ij} \cap \neg R_a^{ik}) \mapsto UNSAT \end{cases} \tag{7}
$$

For clarity, the situations where this conflict may occur is not always considered as conflict but could be considered as a warning or potential conflict, where the end-user

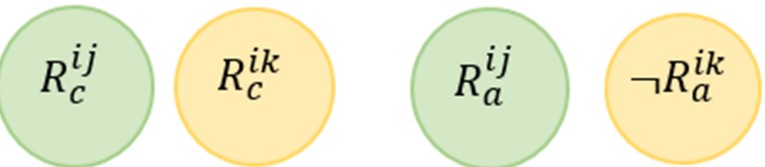

**Figure 10** Condition inconsistency conflict venn diagram representation.

can be notified that she writes something error. According to the end user's intentions or preferences, she can report this as a conflict or warning.

```
R11: IF Temperature >= 30 THEN AirConditioner = True
R12: IF Temperature <  30 THEN AirConditioner = True
```

In the above example, we say that $R^{11} \overset{CIC}{\perp} C^{12}$ according to the definition provided in Eq. (7). In other words, the action of $R^{11}$ supersedes the action of $R^{12}$, but the two rules cannot be simultaneously triggered. For the above rules, there is no valid assignment to satisfy neither $(Temperature >= 30) \cap (Temperature < 30)$, but for $(AirConditioner = True) \cap (AirConditioner = True)$ there is a valid assignment. In this example, the state of AirConditioner is always **ON**, which will increase the energy consumption, therefore the end-users considers it as a conflict.

On the other hand, CIC can be handled as a warning. Consider the following two rules, while there is no valid assignment to satisfy neither $(UseratHome = True) \cap (UseratHome = False)$, but for $(SecurityCamera = True) \cap (SecurityCamera = True)$ there is a valid assignment. In this example, the preference of the end-user may need is the Security Camera to be always running in either he is at home or not, therefore the end-user may notify about this state "Does it legal for him or not".

```
R11: IF UseratHome = True  THEN  SecurityCamera = True
R12: IF UseratHome = False THEN SecurityCamera = True
```

**Race Condition Conflict (RCC)** *When the system has at least one valid situation whereby two different rules' conditions are satisfied (or triggered together after a while) concurrently, and their actions need to update the system state of the shared devices in different ways, letting the system be in a non-specified state. A service rule and a policy constraint may have this conflict as well.*

This conflict is mentioned in *Ibrhim et al. (2020)* as Non-Specified Conflict, called Race Condition in *Celik, Tan & McDaniel (2019)*, and discussed in *Nacci et al. (2018)* for rules that have temporal patterns in their conditions. Figure 11 shows the Venn diagram for the conflict. Formally, this conflict can be described as follows:

$$\overset{RCC}{\perp} := \begin{cases} \exists R^{ij} \text{ and } R^{ik} \in \text{ the same service i where } j \neq k, \\ \forall d1 \in D^{R_c^{ij}} d1 \notin D^{R_c^{ik}}, \exists d2 \in \{D^{R_a^{ij}}, D^{R_a^{ik}}\}, and \\ \exists v1 \in V : (R_c^{ij} \cap R_c^{ik}) \mapsto SAT \text{ and} \\ \qquad (R_a^{ij} \cap R_a^{ik}) \mapsto UNSAT \\ \exists v2 \in V : (R_c^{ij} \cap \neg R_c^{ik}) \mapsto SAT \text{ and} \\ \qquad (R_a^{ij} \cap R_a^{ik}) \mapsto UNSAT \end{cases} \qquad (8)$$

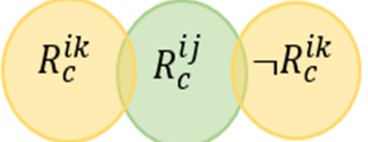 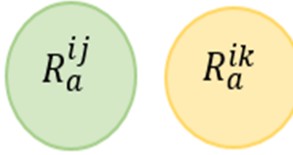

**Figure 11   Race condition conflict venn diagram representation.**

```
R11: IF Occupancy = True THEN AirConditioner = False
R12: IF Temperature > 30 THEN AirConditioner = True
```

In the above example, we say that $R^{11} \overset{RCC}{\perp} C^{12}$ according to the definition provided in Eq. (8). In other words, the rules actions are contradicting but there is a possibility of simultaneously triggering both rules. There is valid assignment that satisfies $(Occupancy = True) \cap \neg (Temperature > 30)$ and a valid assignment that satisfies $(Occupancy = True) \cap (Temperature > 30)$, but for the actions in both cases have no valid assignment that satisfies $(AirConditioner = True) \cap (AirConditioner = False)$.

On the other hand, when RCC conflict occurs with a policy constraint, consider for example the below service rule and policy constraint. Here, we could say that an RCC conflict will occur after a while (e.g., after 5 min) or the rule has a chronological pattern to be executed, and the rule's actions violating the policy constraint which try to make the location more suitable in emergencies.

```
C11: Doors = False AND EmergencySituations = True
R11: AFTER 5 minutes, IF Lecture_started = True THEN LectureDoors = False
```

**Useless Rule Conflict (URC)** *When a rule's conditions are violating all the admissible values determined by a policy constraint (i.e., the semantic rule for a device), we say that the rule is useless.*

This conflict is mentioned in *Ibrhim et al. (2020)*, called Unused Rules in *Vannucchi et al. (2017)*, and known as Policy Violation in *Liang et al. (2016a)*. Figure 12 shows the Venn diagram for the conflict. Formally, this conflict can be described as follows:

$$
\overset{URC}{\perp} := \begin{cases} \exists C^{mn} \text{ and } R^{ij}, \\ \exists d1 \in \{D^{C^{mn}}, D^{R^{ij}_c}\}, and \\ \forall v \in V^{C^{mn}} : (C^{mn} \cap R^{ij}_c) \mapsto UNSAT \end{cases} \tag{9}
$$

Taken separately, a useless rule is considered syntactically valid according to the user preferences. However, when combined with policy constraints, it has no semantic and has no chance of running because its condition is not achievable given the policy constraints.
```
C11: CO2 <= 1000
R11: IF CO2 > 1000 THEN Fan = On
```

With respect to the given example, we say that $R^{11} \overset{URC}{\perp} C^{11}$ according to the definition provided in Eq. (9). In other words, $R^{11}$ should never be triggered because its condition

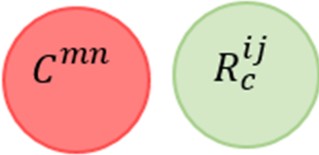

**Figure 12** Useless rule conflict venn diagram representation.



**Figure 13** Range incomplete conflict venn diagram representation.

should never occur. There is no valid assignment to satisfy $(CO2 <= 1000) \cap (CO2 > 1000)$. Here, the policy constraint is defined as the admissible values for the smoke level in a location of interest, and because the user's intentions breakdown this range of safety, his rule is classified as conflict.

**Range Incomplete Conflict (RIC)** *For a service rule and a policy constraint when the rule's conditions satisfy some values of the inadmissible range specified by the constraint.*

This conflict is mentioned in *Ibrhim et al. (2020)* and known as Incorrect Rules in *Vannucchi et al. (2017)*. Figure 13 shows the Venn diagram for the conflict. Formally, this conflict can be described as follows:

$$
\overset{RIC}{\perp} := \begin{cases}
\exists C^{mn} \text{ and } R^{ij}, \\
\exists d1 \in \{D^{C_a^{mn}}, D^{R_c^{ij}}\}, \ \exists d2 \in \{D^{C_c^{mn}}, D^{R_a^{ij}}\}, and \\
\exists v1 \in V^{R_c^{ij}} : (C_a^{mn} \cap R_c^{ij}) \mapsto SAT \ and \\
\qquad\qquad (C_c^{mn} \cap \neg R_a^{ij}) \mapsto UNSAT \\
\exists v2 \in V^{R_c^{ij}} : (C_a^{mn} \cap \neg R_c^{ij}) \mapsto SAT \ and \\
\qquad\qquad (C_c^{mn} \cap \neg R_a^{ij}) \mapsto UNSAT
\end{cases}
\tag{10}
$$

```
C11: IF AirConditioner = True THEN Temperature > 25
R11: IF Temperature > 28 THEN AirConditioner = True
```

With respect to the given example, we say that $R^{11} \overset{RIC}{\perp} C^{11}$ according to the definition provided in Eq. (10). In other words, there is a satisfying assignment for $(Temperature > 25) \cap (Temperature > 28)$, and also for $(Temperature > 25) \cap \neg (Temperature > 28)$.

**Joint-behavior Services Conflict (JBsC)** *When there is at least one valid system situation that triggers two rule conditions belonging to two different users and their joint actions violate a policy constraint.*

A case of this conflict is defined in *Celik, Tan & McDaniel (2019)* under some assumptions. Figure 14 shows the Venn diagram for the conflict. Formally, this conflict

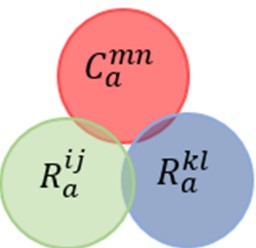

**Figure 14** Joint-behavior services conflict venn diagram representation.

can be described as follows:

$$
\overset{JBsC}{\perp} := \begin{cases} \exists C^{mn}, R^{ij}, and\ R^{kl}, \\ \forall d1 \in D^{R_c^{ij}}, d1 \notin D^{R_c^{kl}}, \forall d2 \in D^{R_a^{ij}},\ d2 \notin D^{R_a^{kl}}, and \\ \exists v1 \in V : (R_c^{ij} \cap R_c^{kl}) \mapsto SAT \\ \qquad (C_c^{mn} \cap (R_a^{ij} \cap R_a^{kl})) \mapsto UNSAT \end{cases} \tag{11}
$$

```
C11: Not(Window AND AirConditioner)
R11: IF S1Occupancy = True THEN Window = True
R21: IF S2Occupancy = True THEN AirConditioner = True
```

As stated in the above example, we say that $R^{11}$ and $R^{21} \overset{JBsC}{\perp} C^{11}$ according to the definition provided in Eq. (11). In other words, there is no satisfying assignment for $\neg(Window = True \cap AirConditioner = True) \cap (Window = True \cap AirConditioner = True)$.

**Device-influence Conflict (DIC)** *For two or more service rules when there is at least one valid system situation triggers the rules and there exists a shared environment entity (e.g., temperature, humidity, light) between them, where their actions affecting this entity in contradicting ways (i.e., affecting the performance of the actions' devices of the other rule).*

This conflict is discussed in *Shah et al. (2019)* and called Opposite-environment-conflict in *Huang, Bouguettaya & Mistry (2020)*. Consider the following situation example in *Huang, Bouguettaya & Mistry (2020)*, suppose a rule that lets the window opened in hot summer while there exists another rule that opens AC for cooling the same location. In this example, opening the window will influence the environment entity "temperature", which in turn dramatically affects the AC performance. This conflict may be considered as a subset of RCC with policy, where if we suppose there is a policy constraint that says," window must be closed when AC is running" to ensure that the system is working in a safe state, this situation can be detected as RCC conflicts.

**Integrity Violation (IV)** *For a service rule, when rule's conditions may be triggered by a third party app or another unauthorized user (e.g., intruder) rule, and its actions directly affect the user' environment.*

This violation is discussed in *Celik, Tan & McDaniel (2019)*. Integrity situations could be occurred due to the deliberate effect of sabotage by using counterfeit components. For example, consider the below service rule. If we suppose that an intruder can deliberately

affect the temperature sensor attached outside the building to increase the temperature value, he can affect the user's indoor environment by letting the air-conditioner turn on.

```
R11: IF outside_SensorTemperature > 23 THEN AC = True
```

**Confidentiality Violation (CV)** *For a user service rule when rule's conditions are satisfied/triggered by the user or system situation and its actions make the information disclosed to a third party app without rule owner consent. This violation is discussed in* Babun et al. (2019).

```
R11: IF end_Meeting = True THEN postFinancialReports_Fb = True
```

For example, consider the above service rule. The rule will post a financial report about a meeting on Facebook when the meeting is ended. Here, the financial reports are considered a piece of private information that must not be shared publicly.

**Privacy Violation (PV)** *For a user service rule, this violation occurs when the rule's conditions are satisfied by a system state, and its actions breakdown the privacy obligations of either the rule owner or others. This violation is discussed in* (Celik et al., 2018; Bastys, Balliu & Sabelfeld, 2018; Tawalbeh et al., 2020).

```
R11: IF OutOffice = True AND Motion = True THEN Camera = True
AND sendPics = True
```

For example, consider the above service rule. The rule will turn on the security camera and send pictures about strangers trying to enter the office, while the user is out of his office and there is a motion near to the office. There are two cases for this rule's actions: one case, if we suppose that these pictures are sent to the rule owner (e.g., his phone) securely, this rule will not consider a violation, since the data is now secure. The other case, if no security on the sending process this rule will violate the privacy of others (e.g., pictures of others may be sniffed by intruders). Babun et al. (2019) suggested when the end-users have a tool by which they can express their privacy preferences, makes the system more secure.

## DISCUSSION

A qualitative comparison for the relevant works resulting from the search process performed in Section 'Survey Methodology' is conducted. The comparison is based on the proposed classification, especially for the automation service level, SLCs, and their related conflicts' types. The results are summarized in Table 2.

### Insights gained from qualitative comparison

Grouping the relevant works according to the proposed conflicts' classification allows us to obtain limitations in the current IoT methods and frameworks of detecting interactions conflicts and insights that will be useful in future IoT automation frameworks as follows:

- In Table 2, the `Conflict Coverage` column is used to represent the percentage between the total number of conflicts in the proposed classification and the number of conflicts covered by the method or tool. Using this percentage, we can determine the capability of the method or tool. For example, approaches proposed in (Hadj et al., 2016; Melissaris,

Ibrhim et al. (2021), *PeerJ Comput. Sci.*, DOI 10.7717/peerj-cs.480

**Table 2  Qualitative comparison between related work based on the explained classification.**

| Type | Approach | Local conflicts | | | | | | | | | Service against policies | | | | Service against services | | | | | | | Conflict Covering (%) |
|---|---|---|---|---|---|---|---|---|---|---|---|---|---|---|---|---|---|---|---|---|---|---|
| | | A | B | C | D | E | F | G | H | I | B | C | J | K | A | B | C | D | E | F | L | |
| | *Resendes, Carreira & Santos (2014)* | | ✓ | | | | | | | | ✓ | | | | | ✓ | | | | | | 15 |
| | *Sun et al. (2014)* | ✓ | ✓ | | ✓ | | | | | | ✓ | | | | ✓ | | | | | | | 25 |
| Conflict | *Magill & Blum (2016)* | | ✓ | | ✓ | ✓ | | | | | | | | | | ✓ | | | | | | 20 |
| | *Sun et al. (2016)* | ✓ | ✓ | | | | ✓ | | | | | | | | ✓ | | | | | ✓ | | 25 |
| | *Mohsin et al. (2016)* | | ✓ | | ✓ | ✓ | | | | | ✓ | | ✓ | ✓ | | ✓ | | | ✓ | | | 40 |
| Classification | *Chi et al. (2018)* | | | | ✓ | | | | | | | | | | | ✓ | ✓ | | | ✓ | | 20 |
| | *Wang et al. (2019)* | ✓ | ✓ | | ✓ | ✓ | ✓ | | | | | | | | | | | | | | | 25 |
| | *Palekar, Fernandes & Roesner (2019)* | | | ✓ | ✓ | | | | | | ✓ | | | | | | | | | | | 15 |
| | *Brackenbury et al. (2019)* | | ✓ | ✓ | ✓ | | | | | | | | | | | ✓ | ✓ | | | | | 25 |
| | *Shah et al. (2019)* | ✓ | ✓ | ✓ | | | ✓ | | | | | | | | | | | | | | | 20 |
| | *Alharithi (2019)* | ✓ | ✓ | | ✓ | | ✓ | | | | | | | | | ✓ | | | | | | 25 |
| | *Chaki, Bouguettaya & Mistry (2020)* | | | | ✓ | | | | | | | | | | ✓ | ✓ | ✓ | | | | | 20 |
| | *Huang, Bouguettaya & Mistry (2020)* | | ✓ | ✓ | ✓ | | ✓ | | | | | | | | | | | | | | | 20 |
| | *Trimananda et al. (2020)* | | | | | | | | | | | | | | | ✓ | ✓ | | | ✓ | | 15 |

**Table 2** (*continued*)

| Type | Approach | Local conflicts | | | | | | | | | Global conflicts – Service against policies | | | | Global conflicts – Service against services | | | | | | | Conflict Covering (%) |
|---|---|---|---|---|---|---|---|---|---|---|---|---|---|---|---|---|---|---|---|---|---|---|
| | | A | B | C | D | E | F | G | H | I | B | C | I | K | A | B | C | D | E | F | L | |
| | *Alhanahnah, Stevens & Bagheri (2020)* | ✓ | ✓ | | ✓ | | | | | | | | | | ✓ | ✓ | | | | | | 25 |
| Conflict — Single User | *Cano, Delaval & Rutten (2014)* | ✓ | ✓ | | ✓ | | | | | | ✓ | | | | | | | | | | | 20 |
| | *Corradini et al. (2015)* | ✓ | ✓ | | ✓ | | | | | | | | ✓ | ✓ | | | | | | | | 25 |
| | *Huang & Cakmak (2015)* | ✓ | ✓ | | | | | | | | | | | | | | | | | | | 15 |
| | *Le Guilly et al. (2016)* | | | | | | | | | | ✓ | ✓ | | | | | | | | | | 10 |
| | *Perumal et al. (2016)* | | ✓ | | ✓ | | | | | | | | | | | | | | | | | 10 |
| | *Peña et al. (2016)* | | | | | | | | | | ✓ | | | | | | | | | | | 5 |
| | *Liang et al. (2016a)* | | | | | | | | | | | ✓ | ✓ | | | | | | | | | 10 |
| | *Vannucchi et al. (2017)* | ✓ | | | | | | | | | | | ✓ | ✓ | | | | | | | | 15 |
| | *Goynugur et al. (2017)* | | ✓ | | | | | | | | | ✓ | | | | | | | | | | 10 |
| | *De Russis & Monge Roffarello (2018)* | ✓ | ✓ | | ✓ | | | | | | | | | | | ✓ | | | | | | 20 |
| | *Celik et al. (2018)* | | | | | | | | | ✓ | | | | | | | | | | | | 5 |
| | *Corno, De Russis & Roffarello (2019)* | ✓ | ✓ | | ✓ | | | | | | | | | | ✓ | | | | | | | 20 |
| | *Al Farooq et al. (2019)* | | | ✓ | ✓ | | | | | | ✓ | | ✓ | | | | ✓ | | | | | 25 |
| | *Manca, Santoro & Corcella (2019)* | | | ✓ | ✓ | | | | | | ✓ | | ✓ | | | | ✓ | | | | | 25 |
| | *Zhang et al. (2019a)* | | | | | | | | | | ✓ | ✓ | ✓ | | | | | | | | | 15 |

Ibrhim et al. (2021), *PeerJ Comput. Sci.*, DOI 10.7717/peerj-cs.480

**Table 2** (*continued*)

Conflict Levels and Types — Local conflicts: A B C D E F G H I; Global conflicts — Service against policies: B C I K; Service against services: A B C D E F L

| Type | Approach | A | B | C | D | E | F | G | H | I | B(p) | C(p) | I(p) | K(p) | A(s) | B(s) | C(s) | D(s) | E(s) | F(s) | L(s) | Conflict Covering (%) |
|---|---|---|---|---|---|---|---|---|---|---|---|---|---|---|---|---|---|---|---|---|---|---|
| | *Krishna et al. (2020)* | | | ✓ | | | | | | | | | | | | | | | | | | 5 |
| | *Munir & Stankovic (2014)* | | | | | | | | | | | | | | ✓ | ✓ | ✓ | | ✓ | | | 20 |
| | *Liang, Hsu & Lin (2014)* | | | | | | | | | | | | | | ✓ | | | | | | | 5 |
| | *Cheng et al. (2014)* | | | | | | | | | | | | | | ✓ | | | | | | | 5 |
| | *Liang et al. (2015)* | | | | | | | | | | ✓ | | | | ✓ | | ✓ | | ✓ | | | 20 |
| | *Yagita, Ishikawa & Honiden (2015)* | | | | | | | | | | | | | | ✓ | | | | ✓ | | | 10 |
| | *Zave, Cheung & Yarosh (2015)* | | | | | | | | | | | | | | ✓ | | | | | | | 5 |
| | *Alfakeeh & Al-Bayatti (2016)* | | | | | | | | | | | | | | ✓ | | | | | | | 5 |
| | *Nandi & Ernst (2016)* | | | | | | | | | | | | | | ✓ | ✓ | | ✓ | | | | 15 |
| Detection | *Hadj et al. (2016)* | | | | | | | | | | | | | | ✓ | | | | | | | 5 |
| (Multiple users) | *Oh et al. (2017)* | | | | | | | | | | | ✓ | | | ✓ | | | | | | | 10 |
| | *Shahi et al. (2017)* | | | | | | | | | | | | | | | | ✓ | | | | | 5 |
| | *Lin et al. (2017)* | | | | | | | | | | | | | | ✓ | | | | | | | 5 |
| | *Celik, McDaniel & Tan (2018)* | | | | | | | | | | | | | | ✓ | ✓ | | ✓ | ✓ | | | 20 |
| | *Celik, Tan & McDaniel (2019)* | | | | | ✓ | ✓ | ✓ | | | | | | | ✓ | ✓ | | | ✓ | | ✓ | 35 |
| | *Melissaris, Shaw & Martonosi (2019)* | | | | | | | | | | | | | ✓ | ✓ | | | | | | | 10 |
| | *Liu et al. (2019)* | | | | ✓ | | | | | | ✓ | | | | | ✓ | | | ✓ | | | 20 |
| | *Nagendra et al. (2019)* | | | | | | | | | | | | | | ✓ | ✓ | ✓ | ✓ | | | | 20 |
| | *Li, Zhang & Shen (2019)* | | | | | | | | | | | | | | ✓ | ✓ | | | | | | 10 |
| | *Xiao et al. (2019)* | | | | | | | | | | ✓ | | | | ✓ | | | | ✓ | | | 15 |
| | *Chen et al. (2019)* | | | | | | | | | | | | | | | | | | ✓ | | | 5 |
| | *Gu et al. (2020)* | | | | | | | | | | | | | | ✓ | ✓ | | | ✓ | | | 15 |
| | *Balliu, Merro & Pasqua (2020)* | | | | ✓ | | ✓ | ✓ | ✓ | | | | | | ✓ | | | | ✓ | | | 30 |
| | *Ibrhim et al. (2020)* | ✓ | ✓ | ✓ | ✓ | ✓ | | | | | ✓ | ✓ | ✓ | ✓ | ✓ | ✓ | ✓ | | ✓ | | | 65 |

**Notes.**

The ✓ marks indicate the conflict detected by the work. Empty cell means the conflict is not detected.

A, Rule Redundancy Conflict; B, Value Inconsistency Conflict; C, Race Condition Conflict.; D, Rule Dependency Conflict; E, Condition Inconsistency Conflict; F, Device-influence Conflict.; G, Integrity Violation; H, Confidentiality Violation; I, Privacy Violation; J, Useless Rule Conflict; K, Range Incomplete Conflict.; L, Joint-behavior Conflict.
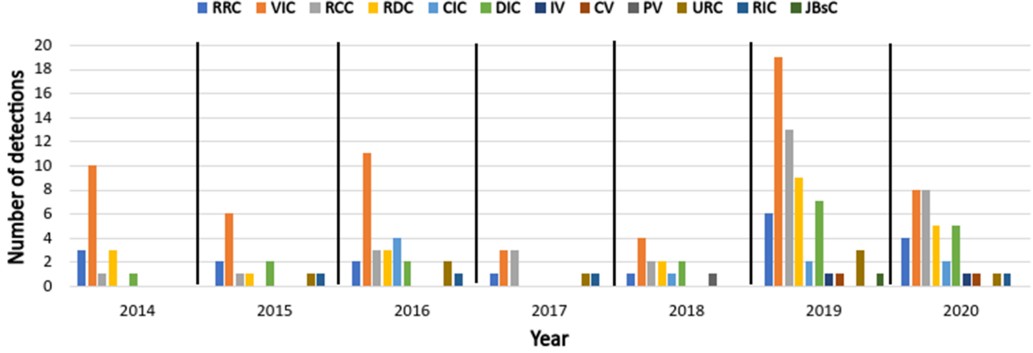

**Figure 15** Classification of conflicts' detection by year according to the reviewed papers.

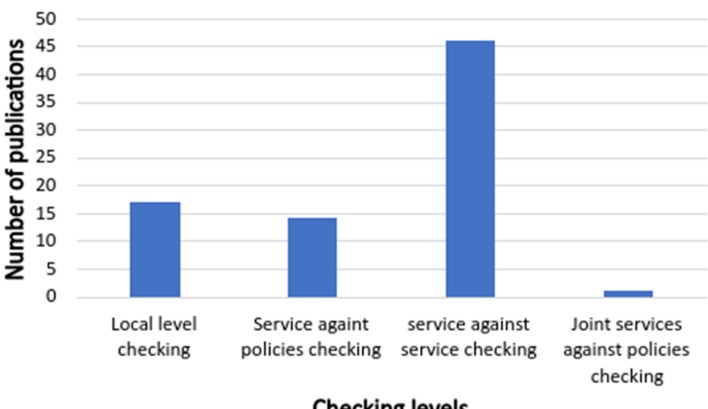

**Figure 16** The number of publications that covers different conflicts' checking levels.

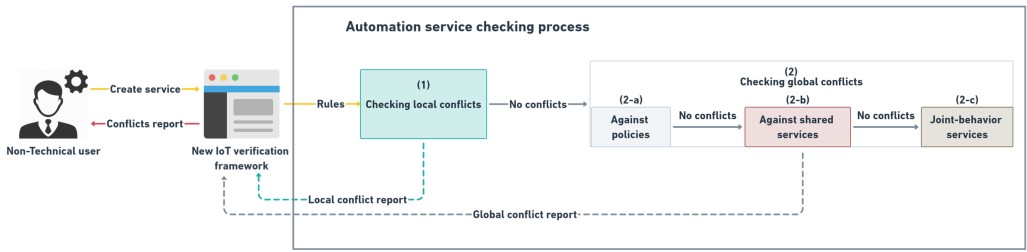

**Figure 17** Automation service checking process according to the proposed conflicts' classification.

*Shaw & Martonosi, 2019*; *Li, Zhang & Shen, 2019*; *Chen et al., 2019*) cover fewer number of conflicts, which affect in the overall system correctness and user safety. On the other hand, methods such as (*Mohsin et al., 2016*; *Celik, Tan & McDaniel, 2019*; *Ibrhim et al., 2020*) although they need more enhancement, they cover a set of mentioned conflict types in the proposed classification.

- A notable gap in previous works is represented in the sparse focusing on the conflicts detection and not covering different conflict types. Figure 15 shows a classification of conflicts' detection by year according to the reviewed papers across the defined period of time. From the figure we could determine the most conflict types that take more researchers attentions such as VIC, RCC, and RDC. On the other side, conflicts such as CIC, IV, CV, PV, RIC, and JBsC have a little attention in the reviewed studies, this is due to different reasons. For instance, in *Vannucchi et al. (2017)*, although it uses the same symbolic verification model checker as in our work proposed in this paper, it fails to detect VIC, RCC, and JBsC conflicts. The reasons for this are, the work in *Vannucchi et al. (2017)* using different ways to detect the conflict, where it includes the invariants to detect the conflicts, and it depends on a single IoT app with its invariants. For instance, RRC in their work is detected only using invariants. Also, IoTGuard in *Celik, Tan & McDaniel (2019)* used another symbolic verification model checker to detect conflicts between multiple IoT apps. However, it fails to detect the JBsC conflict because the unified dynamic model generated by their work required the IoT apps to have a chain of events. According to our work, the JBsC conflict may occur between multiple IoT apps without having this event chain. Similar, the interactions across applications influence discussed in *Chen et al. (2019)* require apps to share an environment entity (temperature) to detect the influence. So, it fails to detect the JBsC conflict when the app does not share devices or environment entities. Approaches proposed in (*Shah et al., 2019*; *Huang, Bouguettaya & Mistry, 2020*) are depending on manual annotations for detecting the conflicts that suffer from rigidity and not suitable for the dynamic and high complexity nature of users intentions. *Corradini et al. (2015)*, Despite having the meaning of constraints over devices, IRON did not consider the conflicts that violate these constraints. (*Li, Zhang & Shen, 2019*), Like our work, the authors of this paper use SMT solver in conflict detection. In our work, we analyze the relationships between IoT apps rules which provide more insights about conflicts. Also, the work in *Li, Zhang & Shen (2019)* is missing the representation of system requirements as constraints which are missing in their work.

- The RVF framework proposed in *Ibrhim et al. (2020)* covers a good set of conflicts as indicated in Table 2. RVF framework provides an incremented and iteratively manner in detecting the intra-automation service and inter-automation services conflicts. Firstly, RVF begins to collect the IoT app's meta-data (e.g., rules, location, execution period, etc.) and converts its rules into an intermediate representation, SMT-Lib v2 language, to be used within SMT solver. After that, it starts checking the IoT app's rules against themselves to detect local conflicts. When such conflicts do not exist, it forwards these rules to the next checking step that is responsible for detecting global conflicts. The global checking within RVF is determined by searching for the overlapping IoT apps in both location and time meta-data. The advantage of using this incremented and iteratively manner is that it adds an acceptable of ensuring that the IoT app will never be executed until all conflicts are detected. For the sake of detecting different levels and types of conflicts, RVF depends on rule relationship analysis using SMT based solver. However,

RVF fails to detect DIC, IV, CV, and PV conflicts because the rule representation used in it does not include the representation of devices' effects on each other.

- Although the state-of-the-art for IoT apps' interactions conflicts detection provide different frameworks and methods, whereas some issues still exist and need to be covered in the future development of IoT verification systems. Among the shortage in the current verification and detection methods are as follows: The works that proposed parser-based methods such as (*Wang et al., 2019*; *Chen et al., 2019*) are built under the assumption that the user inputs are always correct, which does not happen in reality. Also, facing the NLP classification errors. The proposed method in *Xiao et al. (2019)* utilizing different tools to able to analyze the rule structures. In addition to this, these methods suffer from analyzing rules such as *"IF living room temperature is hot THEN cool down the room temperature"*. Such rules that express the user preferences using words instead of Boolean or Integer valued parameters need the using of the string-supported model checker as (*Liang et al., 2016b*; *Abdulla et al., 2015*) which supporting the combination of string constraints and linear integer arithmetic. On the other hand, the methods as (*Magill & Blum, 2016*; *Sun et al., 2014*) either depend on manual annotations of conflicts without checking the validity of these annotations using model checkers or did not provide the definitions of other conflicts. Also, these methods are missing of representing the environment entities that could be affected by different devices; for example, illumination could be affected by a rule that opens a window or another rule that open lights. Adding to this, using single logic for rules interpretations (*Ibrhim et al., 2020*) and others is missing of representing the description of some conflicts like Integrity, Confidentiality, and Privacy. SMT-based methods are missing of representing temporal behaviors rules (i.e., Time-based rules), where rules have some chronological patterns represented by the temporal operators (e.g., FUTURE, PAST, UNTIL, AFTER, BEFORE), instead of representing time as a monotonically increasing variable, where time is considered as a device like a temperature sensor that can send its value when required. Omitting the representation of the temporal operators adds a drawback in the rules' relationships analysis to detect more conflicts. For the sake of covering these issues, we recommend using a hybrid approach that combines model checking with a variety of languages and semantic technologies in developing future IoT-based apps verification frameworks to cover all levels and types of conflicts to guarantee and maximize the safety, security, and correctness of IoT systems.
- Like *Shehata, Eberlein & Fapojuwo (2007)*, the work distinguishes between two types of IoT apps, which are automation service and system policy, as indicated in Table 2, PLCs are ignored in state-of-the-art conflict detection approaches, maybe due to they suppose that the constraints of the policy do not need to be checked because of its domain expert's (administrator, a person who has the authority to write policies) responsibilities. But, in the proposed conflicts' classification, both policies, and automation services are checked against conflicts. Suppose, for example, a location may have more than one administrator, so policy' constraints must check against themselves and other policies.
- Levels of checking process in previous approaches and tools have the limitation of not supporting different automation services checking level. Figure 16 shows that the

conflicts resulting from the interactions between different IoT apps are the main focus in the reviewed papers. As seen in Table 2, `Conflict Classification`, the previous works in IoT apps conflicts' classifications focused mainly on IoT apps rules conflicts either the rules belong to a single app (*Sun et al., 2014*; *Wang et al., 2019*; *Shah et al., 2019*) or multiple apps (*Chi et al., 2018*; *Chaki, Bouguettaya & Mistry, 2020*; *Trimananda et al., 2020*). Little attentions come for violating system policies, these attentions are elucidated in value inconsistency conflict. According to the proposed conflicts' classification which suggests performing the checking and detecting of interactions conflicts in different levels, as shown in Fig. 17. For instance, for the SLCs, there are two levels of checking, Firstly, in (**1**) the rules authored by the non-technical user are checked against themselves to detect local conflicts. Secondly, in the case of no local conflicts, the rules are checked to detect global conflicts (**2**). This level includes three checking steps, which are (**2-a**) to check rules against the system policies in the same location, (**2-b**) to check rules against the shared spatial–temporal services for other users, and (**2-c**) to check each services pair owned by different users against the joint-behavior conflict.

- The proposed conflicts' classification intended use is within smart buildings, smart campus, or any smart consumer of IoT resources. Also, it is based on a general condition-action paradigm with some rules and constraints structures. So, for completing the picture and take more and more advantages from the classification, a standard representation for both automation service rules and system policies which include different IoT platforms (e.g., SmartThings, IFTTT, Zapier, OpenHAB, etc.) is required. The main required feature in this standard representation is the capability of expressing the end-users heterogeneity and complex preferences. With this standard representation, IoT systems developers can make more IoT apps' interactions analysis and take correct decisions for both design time and run time errors and conflicts during the development step of such systems.

- Our survey opens a new research direction in the area of developing custom or domain-specific compilers for the IoT systems verification context. In this context, almost all components (e.g., infrastructures, involved users, applied technologies, services, etc.) within it are characterized by the dynamic nature, which requires the development of compilers that is capable of not only detecting the end-users programming errors but also has the ability to mitigate and resolve conflicts resulting from interacting IoT apps in different levels (i.e., during IoT app design time and within its run time). At last, we suggest using/adding our conflicts' classification in the future formal verification methods or even compilers related to IoT systems, as a part that is responsible for ensuring the correctness and safety of end-users programmed IoT applications.

## CONCLUSIONS

Rule interaction conflicts in IoT apps are a challenging concern. One purpose is sought, which is ensuring the correctness of the interactions between automation services and system policies through detecting as many conflicts as possible. In this survey, IoT systems related conflicts and their levels and types, including the automation services and system

policies, were revealed. The purpose of this survey is to highlight the relevant studies on the IoT apps conflicts definition and detection. This survey proposed a conflicts' classification which aims at filling the gap in relevant studies and considers the first work to collect all types of conflict for IoT systems. The advantages of the proposed conflicts' classification over others are: (1) providing a comprehensive classification of rule-based conflicts in IoT systems with differentiating between the conflicts related to automation services and system policies, (2) supporting multi-level checking, (3) defining the conflicts is based on the formal method and SMT-based checker instead of manual annotations, (4) a step toward developing a custom compiler for IoT-based apps to detect bugs and conflicts in them, and (5) highlighting the uncovered conflicts that should be taken when building accurate IoT verification frameworks.

In light of the conducted review, a domain-specific language is needed to generalize the creation of automation service and system policy in building automation. Another main thing needed is performing an analysis to determine the different ways of solving the conflicts' types stated in the proposed classification. Also, there is a need to reconnoiter the use of hybrid verification and rule formalization tools to guarantee a high level of IoT system safety, security, and correctness. Finally, enhancing the verification model with a level of device interactions and influences knowledge that will help overcome some conflicts.

## ACKNOWLEDGEMENTS

This work is part of a research project entitled: Campus as Mashups Platform for IoT Experimentation (CAMPIE), which is a project of the National Telecommunications Regulatory Authority (NTRA) of Egypt (Project number CFP5/2015/CAMPIE).

### Funding
The authors received no funding for this work.

### Competing Interests
The authors declare there are no competing interests.

### Author Contributions
- Hamada Ibrhim, Hesham Hassan and Emad Nabil conceived and designed the experiments, performed the experiments, analyzed the data, performed the computation work, prepared figures and/or tables, authored or reviewed drafts of the paper, and approved the final draft.

### Data Availability
 We have no raw data or code as this is a literature review.

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
