# Peer review of "A conflicts’ classification for IoT-based services: a comparative survey"

_PeerJ Computer Science, doi:10.7717/peerj-cs.480_

## Round 0.1 · original submission · Major Revisions

Please revise your manuscript addressing issues highlighted by the two anonymous reviewers, especially pay attention to the first reviewer that is reporting several relevant concerns.

Reviewer 1 ·

Basic reporting

The authors provide a review article on BAS conflicts on the level of IFTTT programming.
The paper is mostly well-written and well-structured. Minor typos can be found in the bibliography, e.g. some references have lower-upper-case issues, such as "iot" instead of "IoT".

Experimental design

Clear design. Suitable literature basis in the domain; but lack of related literature. The the paper has some flaws, see (3).

Validity of the findings

The survey results provided by the paper are mostly valid. However, the major flaw of the work is the following: The authors try to describe some essential (mostly semantic) programming issues which are studied in compiler design and formal language verification / language-based security since decades. If the authors would have used the terminology and knowledge from these domains, most of their work would not be new at all, since surveys/review papers in these domains cover them in more detail and with a clearer description.

Some of the findings are moreover trivial, such as condition inconsistency conflict (CTC).

Other conflicts are not stated correctly: especially the "privacy conflict" mixes very different terms such as privacy, confidentiality, danger (safety) and integrity.

The "CIC" condition is not necessarily always wrong and too simplified.

The "Time-based Conflict" is what computer scientists call a race condition and there are tons of work about this ... since decades.

The "Useless Rule Conflict" is also unsuitable. A typical interpreter/compiler would throw a syntax error there. There is no clear semantic provided.

Due to the lack of formal specifications of the author's provided rules and above-mentioned issues, I propose to reject this paper.

Additional comments

I encourage the authors to carefully study work on compiler design, race conditions, side channels in building automation etc. when revising the paper. The whole research domain that you describe would highly benefit if results from other CS domains are considered.

Reviewer 2 ·

Basic reporting

- The writing is clear and understandable though further proofreading is required.
- The font sizes in the figures should be more optimized for print readers.
- Technical keywords are well-introduced.

- However, the title indicates it is specific for BAS while the target papers include generic IoT papers. The authors should clarify the difference in terms of conflict categorization.

Experimental design

This is a comprehensive survey paper on conflict detection for IoT/BAS applications and systems. The reviewer appreciates the authors' contribution here, and gives several comments to improve the study further.

- The process to

Validity of the findings

The authors' other paper (Ibrhim et al. 2020) is highlighted in the last row of Table 2. While it is not unfair to cite their papers, the authors should summarize the basic rationale of why their paper is standing out in terms of their own criteria. The review founds the summary of the technology but not much insight about why it can beat the other technologies.

The reviewer recommends the authors to review the coverage of the papers again especially the arguments that many papers ignore "system policies". System policies are often expressed in other automation services rules. And even further, DepSys paper definitely models system policies as "requirement dependencies".

For IRON paper, it's unfair to argue the proposed tool is "not suitable for non-technical users" given most of the rule detection technologies remain at the algorithm level while the paper proposes more tools for usability.

It's a big burden for readers to validate certain techniques' lack of conflict type coverages. Thus, the reviewer recommends the authors to exemplify several papers to explain why they lack in covering certain types of conflicts in explaining Table 2.

While the authors conclude in the abstract by recommending hybrid model checking, the reasoning is weak in the body. The authors may allocate a couple of paragraphs to explain that.

Additional comments

This paper survey a comprehensive list of papers about conflict types and detection, and then propose categorization. The proposed categories are very helpful for evaluating existing and future frameworks.

---

## Round 0.2 · Minor Revisions

Please pay attention to the reviewers, especially reviewer 3.
Also, improve the paper writing and remove mistakes as suggested.

Reviewer 1 ·

Basic reporting

The authors have addressed my comments.

- decrease size of Fig. 2 so that it matches the size of the other figures, incl. font size

Experimental design

no changes to last review

Validity of the findings

findings are valid; authors have addressed my concerns.

Additional comments

all comments of my previous review have been addressed.

Reviewer 3 ·

Basic reporting

The authors provide a sort of systematic review article to classify conflicts in IoT apps with a main focus onto ECA and IFTTT programming.
The paper is quite well-structured but some minor revisions should be enacted.
Some typos:
line 19: tools that capable -> tools capable
line 29: extend -> extent
line 59: for the sake of this work -> what do you mean with this statement?
line 83: intentions interactions -> please clarify
line 85: work as follows -> work are as follows
figure 1: open -> turn on
lines 134 and 141: remove the initial questions
line 143: through -> done by
lines 187-188 are not clear, please re-write them
line 196 remind -> remaining
line 228 techniques are different -> remove different
line 228 analysis among these differences are: -> analysis. Among these differences THERE are:
line 244 which responsible -> which is responsible
line 406 using Feature -> using feature
line 457 In (...) provided -> In (...) the authors provided
line 472 (...) authors explained -> (...) explained
line 580 the mean reasons -> the main reasons
line 592 may be lead -> may lead
line 598 as following -> as follows
line 816 starting checking -> starts checking
line 817 when not such conflicts exist, it forwarding -> when such conflicts do not exist, it forwards
various lines: lack of -> missing of
line 857 in proposed -> in the proposed

Experimental design

The introduction is quite dispersive...I would focus it more on the content of the paper.
Lines 116-140 should be better elaborated with no repetitions
In the searched dataset you did not consider IEEEXplore, is it considered in Google Scholar?
In the review of current conflict's classification, a critique of the current state of the art is missing, moreover I would have classified somehow the reviewed papers, not simply summarizing them
In figures 4 on the acronyms was not defined, moreover, what is the aim of designing Venn diagrams with no superimpositions? with separate spheres? Moreover, the link between the figures and the preceding equations for each conflict is not clear.
The discussion does not present quantitative evaluations, I would introduce some graphs about the temporal trend of publications, the venues, and the categories of conflicts in time and in venues

Validity of the findings

The survey results provided by the paper are valid but more quantitative results and graphs should be inserted to quantify the conclusions drawn by the authors.

---

## Round 0.3 · accepted · Accept

Please follow the last hints from the reviewer and perform final proofreading.

Reviewer 3 ·

Basic reporting

Some proofreading is still necessary, e.g., do not used "we've" but "we have" in formal writing.
Introduction has been improved.

Experimental design

The drawing of separated Venn diagrams is not yet clear to me.
However, further quantitative investigations have been added.

Validity of the findings

New quantitative results are enough.